# CoT-MT³: CoT-Guided Meta Test-Time Training for Multimodal Reasoning

## Abstract

Large Multimodal Models (LMMs) have achieved remarkable results across various tasks, but they still face challenges in complex multimodal reasoning that is typically performed via chain-of-thought (CoT). Recent studies also start to explore the retrieval-augmented few-shot setting to alleviate this problem. However, existing methods still lack tailored retrieval strategy and effective utilization of demonstrations in complex multimodal reasoning scenarios, resulting in limited reasoning improvements. In this paper, we introduce a novel framework, termed **CoT**-Guided **M**eta **T**est-**T**ime **T**raining (**CoT-MT³**), to enhance LMMs' few-shot multimodal reasoning ability by employing a CoT-guided Weighted Retrieval (CWR) strategy and a Meta Test-Time Training (MT³) paradigm. To provide more relevant demonstrations, CWR employs a retrieval-specific CoT to highlight key information and deep reasoning of the test query for problem-solving. Retrieval is then performed based on the weighted similarity of both the original query and the derived CoT cues. Moreover, to fully leverage retrieved demonstrations, MT³ introduces a context-based meta-learning paradigm by constructing multiple training samples per query with varying context sizes and combinations using few-shot demonstrations. Experiments across three benchmarks show that our CoT-MT³ achieves a significant relative improvement of up to 4.82% on MathVerse and 8.38% on We-Math in the 4-shot setting. Notably, we observe that our CoT-MT³ demonstrates exceptional robustness across different context sizes, highlighting its effectiveness and generalization to few-shot reasoning scenarios.

## 1 Introduction

Large Multimodal Models (LMMs) (Wang et al., 2024b; Liu et al., 2024; Li et al., 2024a) have achieved notable advances in recent years across a wide range of domains. However, they still struggle in solving out-of-distribution questions (Zhang et al., 2024c; Han et al., 2023), especially in complex multimodal reasoning (Zhang et al., 2024a; Wang et al., 2024a) that is typically performed via chain-of-thought (CoT). To alleviate this issue, recent studies (Wang et al., 2023; Zuo et al., 2025; Muennighoff et al., 2025; Snell et al., 2024b; Akyürek et al., 2024) explore test-time scaling strategies, which improves model performance by incorporating additional inference-time compute or task-specific information during inference. Among these strategies, retrieval-augmented methods (Dong et al., 2024; Hübotter et al., 2024) have emerged as a promising direction, which retrieve few-shot demonstrations (also including CoT) at test time to boost the performance of LMMs.

However, these retrieval-augmented approaches remain underexplored in complex reasoning scenarios, which still fall short in achieving accurate retrieval and fully leveraging the retrieved few-shot demonstrations, thus yielding limited improvements. Firstly, existing retrieval mechanisms (Liu et al., 2023; Dong et al., 2024; Tan et al., 2024) primarily rely on question-based similarity between the test query and candidate questions, while overlooking the deep reasoning behind the test query (i.e, the relevant mathematical principles and possible solution strategies). As shown in Figure 1, retrieval solely based on the question leads to the selection of reasoning-level inconsistent demonstrations, and thus fails to provide sufficient support for problem solving. This bias significantly hinders performance on tasks demanding complex multi-step reasoning (Fu et al., 2022).

Furthermore, the complexity of multimodal data also poses significant challenges in leveraging the retrieved few-shot demonstrations. There are two main strategies to leverage these demonstrations:

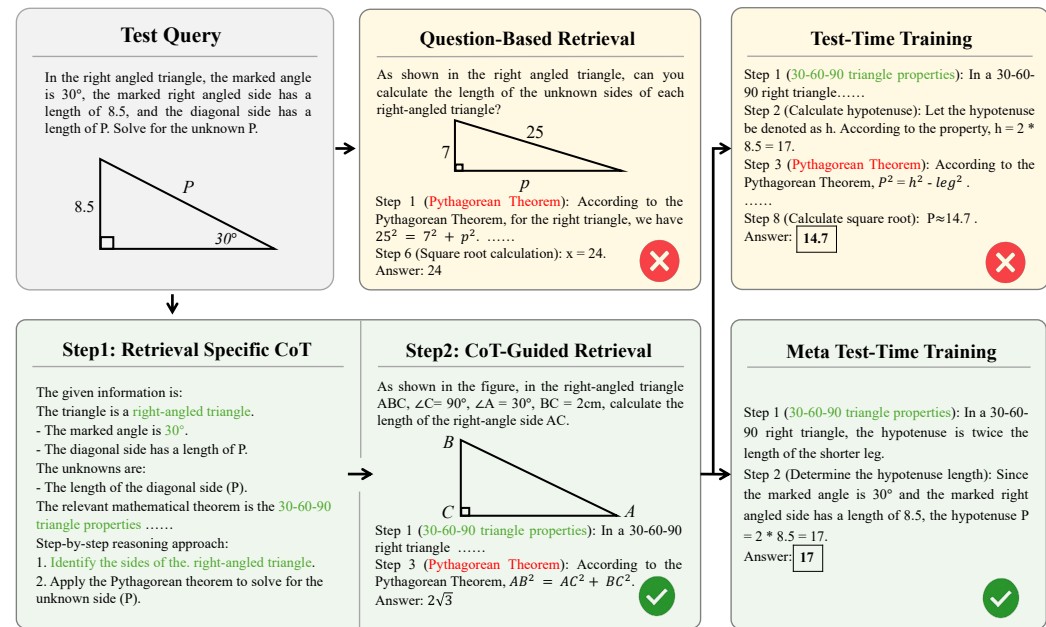

Figure 1: Comparison between different retrieval strategies and training paradigms. It can be seen that CoT-guided retrieval can more effectively search demonstrations with higher similarity in both problem formulation and problem-solving approaches than question-based retrieval. Moreover, simple fine-tuning approach tends to overfit to the retrieval demonstration and copy their reasoning patterns directly, which otherwise can be alleviated by meta test-time training.

1) In-Context Learning (ICL) that provides demonstrations in prompts for reference (Liu et al., 2023; Dong et al., 2024; Tan et al., 2024; Jiang et al., 2024; Qin et al., 2023), and 2) Test-Time Training (TTT) that fine-tunes the model with these lightweight demonstrations at test time (Hardt & Sun, 2024; Hübotter et al., 2024). However, ICL methods struggle to understand complex multimodal prompts with multiple interleaved images and texts. As the number of demonstrations increases, ICL methods even actually harm the reasoning performance (Qin et al., 2024; Liu et al., 2023). Meanwhile, TTT methods tend to overfit to the limited number of demonstrations, causing the model to copy the pattern of the demonstration directly, which leads to incorrect answers (Hübotter et al., 2024). Overall, both groups of retrieval-augmented approaches (i.e., ICL and TTT) fail to fully leverage the retrieved few-shot demonstrations in boosting the reasoning ability of LMMs.

To address the above limitations, we propose a novel framework, termed **CoT**-Guided **M**eta **T**est-Time **T**raining (**CoT-MT**[3]), to enhance LMMs' complex multimodal reasoning performance during test time. The proposed framework consists of two key components: a CoT-guided Weighted Retrieval (**CWR**) strategy and a Meta Test-Time Training (**MT**[3]) paradigm. As shown in Figure 1, the CWR strategy improves retrieval accuracy through two modules: retrieval-specific CoT and CoT-integrated weighted retrieval. The retrieval-specific CoT decomposes the reasoning process into multiple predefined sub-tasks, guiding the original LMM to highlight key problem information and task-specific knowledge for solving problems, such as relevant mathematical theorems, as illustrated by the green text on the left side of Figure 1. The CoT-integrated weighted retrieval strategy then selects target demonstrations based on the weighted score of question similarity and reasoning similarity (computed between the CoT output and the derived CoT cues).

Built upon CWR, our MT[3] paradigm introduces a context-based meta-learning paradigm designed to improve LMMs' reasoning ability at test time. Rather than directly fine-tuning on the fixed set of retrieved demonstrations, MT[3] constructs a series of few-shot training samples with varying context sizes and diverse combinations. Each demonstration is treated as the target in turn, while the remaining demonstrations are selected, mixed up and utilized to form its prompt context. This training process encourages the model to learn how to recognize useful information under diverse multimodal prompt conditions. In this way, our method fully leverages the potential of the retrieved demonstrations in mete-learning way to achieve robust reasoning of LMMs at test time.

Our contributions are summarized as follows: **1)** We propose CoT-Guided Weighted Retrieval (CWR) strategy that combines retrieval-specific CoT with a CoT-integrated weighted mechanism to retrieve demonstrations with higher accuracy. **2)** We introduce $MT^3$, a context-based meta-learning paradigm that improves the model's robustness across varying few-shot settings and facilitates effective reasoning at test time. **3)** Extensive experiments show that the proposed CoT-$MT^3$ significantly improves LMMs' complex reasoning ability, and outperforms other competing methods across most settings, demonstrating its effectiveness in retrieval-augmented reasoning scenarios.

## 2 RELATED WORK

**Multimodal Reasoning.** With the growing attention on multimodal reasoning, a variety of methods (Peng et al., 2024; Shi et al., 2024; Gao et al., 2023) and benchmarks (Zhang et al., 2024a; Lu et al., 2024; Qiao et al., 2024; Wang et al., 2024a; 2025a) have been introduced, contributing to advancements in the field. Most existing approaches (Shi et al., 2024; Li et al., 2024b) rely on fine-tuning LMMs using large-scale multimodal datasets to enhance their reasoning abilities. Due to the scarcity of high-quality multimodal data, fine-tuning on synthetic data (Zhang et al., 2024b; Gao et al., 2023) has emerged as a widely adopted strategy, yielding some improvements in model performance. Recently, test-time scaling techniques have gained traction as an alternative approach to enhance reasoning performance (Muennighoff et al., 2025; Guan et al., 2025; Ye et al., 2025; Snell et al., 2024a; Dong et al., 2024). Among them, retrieval-augmented approaches have demonstrated effectiveness (Dong et al., 2024; Liu et al., 2023; Tan et al., 2024). However, their application in complex multimodal reasoning remains largely unexplored. Developing techniques that can effectively leverage retrieved few-shot demonstrations and adapt LMMs to complex multimodal reasoning tasks during inference remains a critical challenge.

**Test-Time Training.** Test-Time Training (TTT) (Sun et al., 2020; Hardt & Sun, 2024) is a general approach for enhancing model performance when training and test data come from different distributions. Recent works on TTT have extended this paradigm to LLMs (Hardt & Sun, 2024; Akyürek et al., 2024; Wang et al., 2024c; Hübotter et al., 2024) by fine-tuning on retrieved demonstrations, demonstrating its effectiveness on novel tasks. TTT-NN (Hardt & Sun, 2024) improves language modeling task performance by fine-tuning top-$N$ nearest neighbors retrieved from each test query. Similarly, TTT-ICL (Akyürek et al., 2024) constructs context-based demonstrations according to few-shot data for fine-tuning, achieving strong results on the ARC Challenge. However, TTT hasn't been explored in complex multimodal reasoning scenarios, particularly in terms of demonstration multimodal retrieval and effectively reasoning under few-shot conditions.

**Chain-of-Thought Reasoning.** Chain-of-Thought (CoT) (Wang et al., 2025b; Wei et al., 2022; Chen et al., 2025) has significantly advanced LMMs' reasoning abilities, leading to notable progress in solving multi-step reasoning tasks. Apart from fine-tuning approaches, existing works explicitly generate intermediate steps or decompose the problem into manageable subproblems, thereby enabling models to tackle complex tasks in a interpretable manner (Zhang et al., 2023; Zheng et al., 2023; Sun et al., 2025). Recent works (Qin et al., 2023; Trivedi et al., 2022) also propose to leverage the model's initial CoT outputs to retrieve relevant demonstrations and enhance downstream tasks through retrieval-augmented methods. However, these methods overlook the explicit optimization of the CoT reasoning process for retrieval purpose. In this work, we propose a retrieval-specific CoT that highlights key information to support tailored demonstration retrieval.

## 3 METHODOLOGY

### 3.1 PRELIMINARY

In the retrieval-augmented few-shot setting, given a test query $q_t = \{i_q, t_q\}$, where $i_q$ denotes the image and $t_q$ denotes the question text, along with a demonstration pool $D$, the first step is to retrieve the most relevant $m$ demonstrations from $D$. This is achieved via a similarity function $S(x_q, x)$ that ranks each candidate $x \in D$ based on its relevance to the test query $x_q$:

$$X = \{x_1, x_2, \ldots, x_m\} = \text{top-}m(D, S(x_q, \cdot)), \tag{1}$$

where each retrieved demonstration $x_i = \{q_i, r_i\}$ consists of a question $q_i$ and a corresponding response $r_i$, and the function top-$m(D, S)$ denotes the most relevant $m$ demonstrations from $D$ according to the similarity function $S(x_q, \cdot)$.

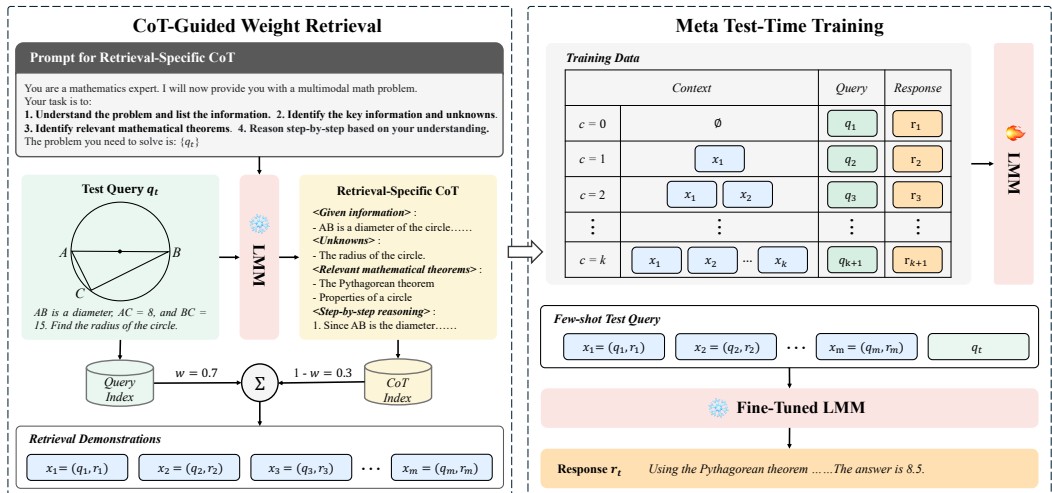

Figure 2: **Overview architecture of our proposed CoT-MT**[3]. It consists of two novel components:
**(1) CoT-Guided Weight Retrieval:** Given the test query $q_t$, the original LMM first generates a retrieval-specific CoT that captures task-specific information. This information combined with the test query is utilized in a weighted retrieval mechanism to retrieve top-$m$ relevant demonstrations.
**(2) Meta Test-Time Training:** Built upon the retrieval demonstrations $\{x_1, x_2, ...x_m\}$, the model is fine-tuned using a series of few-shot training samples. For each query with question $q_i$, multiple training samples ranging from 0-shot to $k$-shot are constructed by random sampling different subsets of the retrieved demonstrations. During inference, the fine-tuned LMM leverages the test query with $k$-shot retrieved demonstrations to obtain the final response $r_t$.

The objective of retrieval-augmented few-shot learning is to: (1) optimize the selection of relevant demonstrations and (2) maximize the model's ability to generate accurate predictions conditioned on the selected demonstrations. This can be formulated as:

$$\max_{X \subset D} P(r_q \mid x_q, X), \tag{2}$$

where $P(r_q \mid x_q, X)$ denotes the probability of generating the response $r_q$ for the query $x_q$, conditioned on the retrieved demonstrations $X$.

## 3.2 OVERALL ARCHITECTURE

Our goal is to enhance LMMs' reasoning performance under retrieval-augmented few-shot setting. As illustrated in Figure 2, the proposed framework comprises two key components: CoT-guided Weighted Retrieval (CWR) and Meta Test-Time Training (MT[3]). CWR improves retrieval quality by employing a retrieval-specific CoT that decomposes the initial reasoning process into multiple sub-tasks, guiding the model to highlight key information and task-specific knowledge. A CoT-integrated weighted retrieval mechanism is then employed to select demonstrations by combining question-based similarity and reasoning-based similarity. In the test-time training stage, we propose MT[3], a context-based meta-learning paradigm to improve LMMs' reasoning ability at test time. Rather than simple fine-tuning, MT[3] constructs few-shot training samples with varying context sizes and combinations, encouraging the model to learn how to recognize valuable information and achieve effective reasoning from multimodal context. We describe the details of each module below.

## 3.3 COT-GUIDED WEIGHTED RETRIEVAL

### 3.3.1 RETRIEVAL-SPECIFIC COT

In multimodal reasoning tasks, retrieving highly relevant demonstrations requires precise understanding and deep analysis of the problem content. A natural solution is to leverage the model's preliminary Chain-of-Thought (CoT) reasoning output as auxiliary information to improve the retrieval precision (Dong et al., 2024; Qin et al., 2023). However, basic CoT prompting strategies (e.g., "Let's think step by step") focus solely on solving the target problem, making it difficult to extract the key reasoning information for effective retrieval. The mismatch between CoT objectives

and retrieval-specific reasoning demands causes basic CoT prompting to fall short in addressing retrieval-specific requirements.

To address this issue, we propose a retrieval-specific CoT, which structures the model's initial reasoning into a sequence of predefined sub-tasks aimed at uncovering the deep reasoning behind the test query. As illustrated in Figure 3, retrieval-specific CoT decomposes the reasoning into four key stages: understanding and listing the problem statement, identifying key information and unknowns, identifying relevant mathematical theorems, and performing step-by-step reasoning based on above understanding. This structured approach simplifies reasoning by breaking the problem into manageable components while highlighting retrieval-critical elements.

> **Retrieval-Specific COT**
>
> You are a mathematics expert. I will now provide you with a multimodal math problem.
> Your task is to:
> **1. Understand the problem and list the information**:
> - List all the given information and elements from the text and the image in the problem.
> **2. Identify the key information and unknowns**:
> - Extract critical information for solving the problem and highlight any unknowns that need to be determined.
> **3. Identify relevant mathematical theorems**:
> - Identify the relevant mathematical theorems that form the basis for solving the problem.
> **4. Reason step-by-step based on your understanding**
> - Based on your understanding of the problem, attempt to break it down into logical steps and provide a step-by-step reasoning approach to solving the problem.
> The problem you need to solve is:
> <image>
> <question>

Figure 3: Illustration of retrieval-specific CoT for multimodal mathematical reasoning, which decomposes the reasoning process into predefined sub-tasks that guide the model to highlight task-relevant information.

In contrast to basic CoT prompting which primarily generates calculations steps to reach the final answer, our approach emphasizes both the model's understanding and reasoning patterns of the problem. By explicitly guiding the model to construct a retrieval-specific representation of the problem, retrieval-specific CoT ultimately improves the retrieval precision. Moreover, the structure of retrieval-specific CoT can be flexibly adapted to other domains (e.g., physics) to better capture domain-specific knowledge.

### 3.3.2 COT-INTEGRATED WEIGHTED RETRIEVAL

After obtaining the retrieval-specific CoT output, we aim to incorporate both the question content and the generated reasoning information into the retrieval process. However, the question's visual and textual descriptions already occupy substantial token space, while the generated CoT reasoning steps tend to be also detailed. As a result, embedding all components into a unified representation leads to degraded retrieval quality. Furthermore, as different tasks emphasize question and reasoning to different extents, a task-adaptive weighted mechanism is required to balance their contributions.

To this end, we adopt a weighted retrieval strategy that separately computes similarities from question and reasoning, and then dynamically adjusts their influence during retrieval. Specifically, given a test query $x_q$ and the generated retrieval-specific CoT $rs_q$, we compute two types of similarity: question-based similarity and reasoning-based similarity. Let $\text{sim}(\cdot, \cdot)$ denote a similarity function. The question-based similarity $s_q$ is computed between the encoded features of the test query and the candidate demonstration $x_i = \{i_i, t_i\}$:

$$s_q = \text{sim}\left(f(x_q), f(x_i)\right) \tag{3}$$

where $f(x_q)$ and $f(x_i)$ denote the joint multimodal feature embedding of the test query and the candidate demonstration, respectively. The reasoning-based similarity $s_r$ is calculated using the retrieval-specific CoT output $rs_q$ and the response $r_i$ of the candidate demonstration:

$$s_r = \text{sim}(f(rs_q), f(r_i)). \tag{4}$$

To balance their contributions, we define a weighted similarity:

$$s = w \cdot s_q + (1 - w) \cdot s_r, \tag{5}$$

where $w \in [0, 1]$ is a hyperparameter controlling the trade-off between question-based similarity and reasoning-based similarity. This weighted design provides fine-grained control over retrieval relevance, leading to more accurate selection of demonstrations.

### 3.4 META TEST-TIME TRAINING

Although retrieval-augmented methods provide relevant demonstrations at test time, effectively utilizing them to improve multimodal reasoning ability remains challenging. To mitigate this limitation, we propose Meta Test-Time Training (MT$^3$), a context-based meta-learning paradigm. To fully

leverage the retrieved demonstrations, $MT^3$ fine-tunes the model in a meta-learning paradigm using a series of few-shot training instances with varying context sizes and combinations. This enables LMMs to efficiently acquire domain-specific reasoning capabilities at test time, thereby enhancing the overall performance on complex multimodal reasoning tasks.

**Training Set Construction.** As illustrated in Figure 2, we construct a series of few-shot samples for meta test-time training by varying the number and combination of context demonstrations per query. Specifically, given the retrieved demonstration set $X = \{x_1, x_2, \ldots, x_m\}$, where each $x_i = \{q_i, r_i\}$, we generate $k+1$ training samples for each target $x_i \in X$. Each sample is assigned a unique context size from the set $\{0, 1, \ldots, k\}$, where $k \leq m-1$ is a predefined maximum context size. For each context size $c$, the prompt $P_i^{(c)}$ for target $x_i$ is formed by randomly sampling $c$ demonstrations from the remaining set:

$$\forall x_i \in X, \ \forall c \in \{0, 1, \ldots, k\}, P_i^{(c)} \subset X \setminus \{x_i\}, \quad |P_i^{(c)}| = c. \tag{6}$$

Notably, for edge cases such as $c = 0$, there only exist $m$ unique samples. Therefore, we uniformly sample $m$ training samples for each context size to ensure balanced training across context sizes. Additionally, we ensure that each demonstration is used equally as both the target and part of the context, promoting balanced participation and reducing overfitting to specific demonstrations.

**Meta Test-Time Training and Inference.** At test time, we adapt the model using pre-constructed samples generated from retrieved demonstrations. Each training sample consists of a target question paired with a context with size $c$. The training objective is defined as:

$$\mathcal{L}(x_i, P_i^{(c)}) = -\log P(r_i \mid q_i, P_i^{(c)}), \tag{7}$$

where $P(r_i \mid q_i, P_i^{(c)})$ denotes the probability of generating the correct response $r_i$ for the target question $q_i$, conditioned on its associated context $P_i^{(c)}$. The diversity of multimodal prompt conditions in the few-shot training samples enables the model to learn how to identify useful information and enhance reasoning capabilities during meta-training.

Final inference is performed by the fine-tuned model. Following Flamingo (Alayrac et al., 2022), we construct few-shot test query by concatenating original test query and all retrieved demonstrations, sorted by descending similarity to the test query. The fine-tuned model then performs more accurate and robust reasoning based on the retrieval-augmented multimodal context.

# 4 EXPERIMENTS

## 4.1 EXPERIMENTAL SETUP

**Benchmarks.** We focus on the multimodal mathematical reasoning, which serves as one of most challenging tasks for multimodal reasoning. Our method is evaluated on three multimodal mathematical reasoning benchmarks: MathVerse (Zhang et al., 2024a), MathVista (Lu et al., 2024), and We-Math (Qiao et al., 2024), using the testmini sets of each. For MathVerse, we focus on four multimodal subsets: Text Dominant (TD), Text Lite (TL), Vision Dominant (VD), and Vision Intensive (VI), which all involve both textual and visual inputs. We exclude the Text Only and Vision Only subsets to ensure that test queries and retrieved demonstrations share the same input modalities. For MathVista, we evaluate on the Geometric Problem Solving (GPS) subset, and for We-Math, we use the full set. A more detailed description of these benchmarks is provided in Appendix B.1.

**Baselines.** Our method is compared against a range of baseline methods under 2, 4 and 6-shot settings: **(1) Zero-shot:** direct inference without any demonstrations. **(2) Random:** ICL with randomly sampled demonstrations from the candidate pool. **(3) RICES:** retrieval-based in-context example selection (Alayrac et al., 2022), which retrieves demonstrations using visual similarity to the query. **(4) QBICL:** ICL using question-based retrieval, incorporating both the image and question text in the similarity computation. **(5) TTT-NN:** TTT on nearest retrieved demonstrations, following the setup in Hardt & Sun (2024). **(6) TTT-ICL:** TTT using in-context demonstrations, where we follow the leave-one-out construction strategy in Akyürek et al. (2024). Note that both TTT-NN and TTT-ICL adopt question-based retrieval to ensure consistency in comparison.

**Implementation Details.** For the retrieval component, we employ Vista (Zhou et al., 2024), a multimodal hybrid retriever capable of processing long input sequences. All retrieval tasks are

Table 1: Comparative results on MathVerse under 2-shot, 4-shot, and 6-shot settings. Accuracy (%) is used as the evaluation metric. The best score for each setting is **bolded**. All compared methods employ the same backbone Qwen2-VL-7B.

| Methods | TD | | | TL | | | VI | | | VD | | | Avg | | |
|---|---|---|---|---|---|---|---|---|---|---|---|---|---|---|---|
| | 2-shot | 4-shot | 6-shot | 2-shot | 4-shot | 6-shot | 2-shot | 4-shot | 6-shot | 2-shot | 4-shot | 6-shot | 2-shot | 4-shot | 6-shot |
| Zero-shot | 32.49 | 32.49 | 32.49 | 27.41 | 27.41 | 27.41 | 23.73 | 23.73 | 23.73 | 24.49 | 24.49 | 24.49 | 27.03 | 27.03 | 27.03 |
| Random | 31.35 | 30.33 | 31.60 | 27.03 | 25.89 | 25.63 | 22.59 | 22.34 | 25.00 | 23.22 | 24.37 | 25.13 | 26.05 | 25.73 | 26.84 |
| RICES | 33.50 | 36.17 | 34.39 | 28.30 | 29.57 | 28.55 | 24.37 | 26.65 | 25.76 | 22.59 | 22.72 | 24.62 | 27.19 | 28.78 | 28.33 |
| QBICL | 36.80 | 36.80 | 37.69 | 27.92 | 29.19 | 27.66 | 24.49 | 25.12 | 25.76 | 23.98 | 25.63 | 23.60 | 28.30 | 29.19 | 28.68 |
| TTT-NN | **37.06** | 38.96 | 36.80 | 28.55 | 29.19 | 29.19 | 24.75 | 24.87 | 27.03 | 24.11 | 26.40 | 26.40 | 28.62 | 29.86 | 29.86 |
| TTT-ICL | **37.06** | 37.06 | 38.07 | 28.93 | 31.47 | 27.92 | 25.00 | 27.53 | 25.63 | **25.76** | 26.52 | 23.35 | 29.19 | 30.65 | 28.74 |
| CoT-MT$^3$ | 34.77 | **40.36** | **39.97** | **30.46** | **31.60** | **33.88** | **27.28** | **27.66** | **27.16** | 24.87 | **27.79** | **27.79** | **29.35** | **31.85** | **32.20** |

Table 2: Comparative results on MathVista (GPS subset) under 2-shot, 4-shot, and 6-shot settings. Accuracy (%) is used as the evaluation metric. The best score for each setting is **bolded**.

| Shots | Zero-shot | Random | RICES | QBICL | TTT-NN | TTT-ICL | CoT-MT$^3$ |
|---|---|---|---|---|---|---|---|
| 2-shot | 46.15 | 42.31 | 42.31 | 48.56 | 49.52 | 52.40 | **57.21** |
| 4-shot | 46.15 | 40.87 | 49.04 | 46.63 | 54.81 | 56.25 | **60.58** |
| 6-shot | 46.15 | 39.42 | 50.96 | 45.67 | 55.77 | 53.37 | **59.62** |

conducted from the MultiMath-300K (Peng et al., 2024) corpus, a high-quality multimodal bilingual dataset with detailed CoT annotations. To preserve linguistic consistency and semantic alignment, we retrieve demonstrations in the corresponding language of the input query. We employ LoRA (Hu et al., 2022) fine-tuning with a rank of 8 and a scaling factor $\alpha = 16$. The model is optimized using the Adam (Kingma & Ba, 2014) optimizer with a learning rate of 0.0002 and a weight decay of 0.1. The $w$ for CWR is set to 0.7 and the $k$ for MT$^3$ is defined as $\lfloor m/2 \rfloor$, where $m$ represents the number of retrieved demonstrations. All experiments are conducted on 4 NVIDIA A800 GPUs.

## 4.2 MAIN RESULTS

### 4.2.1 RESULTS ON MATHVERSE AND MATHVISTA

**Effectiveness of CoT-MT$^3$.** As shown in Tables 1 & 2, our CoT-MT$^3$ consistently achieves the best or near-best performance across all subsets and few-shot settings. For example, on the TD subset, our CoT-MT$^3$ outperforms TTT-ICL by 3.30% and zero-shot baseline by up to 7.87% under the 4-shot setting. Similarly, on the GPS subset, it exceeds TTT-ICL by 4.33% and surpasses zero-shot baseline by up to 14.43%. Across all 18 evaluation settings ((5 subsets + 1 avg) × 3 few-shot settings), our CoT-MT$^3$ achieves the highest score in 16 out of 18 settings (including the average evaluation settings). These results highlight the strong generalization ability of our CoT-MT$^3$, establishing it as a effective framework for retrieval-augmented multimodal reasoning.

**Potential of TTT-Based Methods.** TTT-based methods exhibit strong potential in retrieval-augmented reasoning tasks. Among them, TTT-NN that performs direct fine-tuning on retrieved demonstrations, shows consistent gains as the number of retrieved demonstrations increases. However, it only employs simple fine-tuning paradigm and thus shows only limited improvement, in comparison with TTT-ICL and CoT-MT$^3$ which incorporate retrieved demonstrations as context.

Furthermore, although TTT-ICL generally outperforms TTT-NN in the 2- and 4-shot settings, its performance declines in the 6-shot scenario. This degradation is likely due to a mismatch between the context length used during training and those encountered at test time. Specifically, the leave-one-out construction strategy of TTT-ICL treats each retrieved demonstration as a test instance, with the rest forming its context, leading to shorter training inputs. Such a mismatch may hinder the model's adaptation to longer and more complex test-time prompts.

In comparison, our proposed CoT-MT$^3$ achieves consistently strong performance across all few-shot configurations. This robustness can be attributed to its meta-learning paradigm, which enhances the model's ability to generalize by adapting to varying multimodal prompt conditions.

Table 3: Comparative results on We-Math. Five evaluation metrics are reported: IK (insufficient knowledge), IG (inadequate generalization), CM (complete mastery), RM (rote memorization), and Avg (loose overall average scores). The best score for each setting is **bolded**.

| Methods | IK (↓) | | | IG (↓) | | | CM (↑) | | | RM (↓) | | | Avg (↑) | | |
|---|---|---|---|---|---|---|---|---|---|---|---|---|---|---|---|
| | 2-shot | 4-shot | 6-shot | 2-shot | 4-shot | 6-shot | 2-shot | 4-shot | 6-shot | 2-shot | 4-shot | 6-shot | 2-shot | 4-shot | 6-shot |
| Zero-shot | 56.19 | 56.19 | 56.19 | 12.95 | 12.95 | 12.95 | 25.14 | 25.14 | 25.14 | 18.52 | 18.52 | 18.52 | 31.62 | 31.62 | 31.62 |
| Random | 57.90 | 61.90 | 57.33 | 9.90 | 10.10 | 11.43 | 26.86 | 24.38 | 25.90 | 16.57 | 12.93 | 17.07 | 31.81 | 29.43 | 31.62 |
| RICES | 60.19 | 58.86 | 56.00 | 11.43 | 9.14 | 9.71 | 22.67 | 26.10 | 30.48 | 20.13 | 18.45 | 11.11 | 28.38 | 30.67 | 35.33 |
| QBICL | 56.76 | 60.00 | 55.81 | 7.81 | **8.00** | 10.48 | 29.33 | 26.48 | 27.81 | 17.20 | 17.26 | 17.51 | 33.24 | 30.48 | 33.05 |
| TTT-NN | **54.67** | 53.52 | 56.76 | 10.86 | 12.29 | 10.67 | 29.14 | 29.52 | 27.62 | 15.47 | 13.89 | 15.20 | 34.57 | 35.62 | 32.95 |
| TTT-ICL | 58.29 | 52.76 | 55.62 | **7.62** | 10.67 | **8.95** | 27.62 | 30.48 | 31.43 | 18.99 | 16.67 | **11.29** | 31.43 | 35.81 | 35.90 |
| CoT-MT[3] | 55.81 | **49.90** | **53.52** | 9.14 | 10.67 | **8.95** | **30.48** | **34.67** | **32.19** | 13.04 | 12.08 | 14.21 | **35.05** | **40.00** | **36.67** |

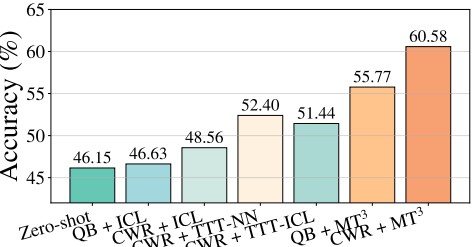

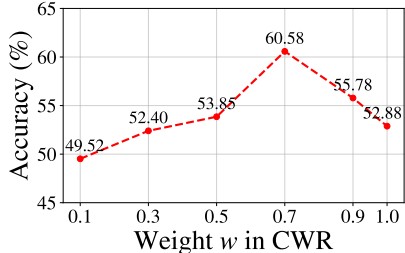

Figure 4: Ablation results for different components of CoT-MT[3] on MathVista (GPS).

Figure 5: Ablation results for different $w$ values in the CWR strategy on MathVista (GPS).

**Validity of Reasoning Information for Retrieval.** RICES relies solely on visual input and performs well in vision-intensive subsets but struggles in text-centric subsets (e.g., MathVerse TD). In contrast, QBICL considering both textual and visual components, yields more balanced performance across different subsets, consistent with findings from prior work (Qin et al., 2024). Built on this, our CoT-MT[3] further integrates reasoning information into the retrieval, guiding demonstration selection based not only on question content but also on underlying reasoning semantics. According to Tables 1 & 2, the reasoning-guided retrieval proves particularly effectiveness for complex multi-step reasoning problems. Overall, it suggests that progressively enriching the retrieval information (from visual, to multimodal, to CoT-guided), substantially improves the relevance of demonstrations.

### 4.2.2 MORE RESULTS ON WE-MATH

Table 3 presents the evaluation results on We-Math across five diagnostic metrics. Our CoT-MT[3] consistently achieves the highest average score across all few-shot settings, with a peak value of 40.00% in the 4-shot setting, significantly outperforming all baselines. In particular, it consistently achieves the highest scores in CM across all few-shot settings, reflecting the model's improved ability to generate complete and well-reasoned solutions. Moreover, our CoT-MT[3] maintains competitive performance in IK and RM, indicative of reduced fundamental misunderstandings, suggesting that the combination of CWR and MT[3] strengthens the model's overall reasoning capacity while enhancing its conceptual clarity. While TTT-ICL and TTT-NN demonstrate strong results in selected metrics (e.g., TTT-ICL achieves the best IG score at 2-shot), they suffer from less consistent performance across few-shot settings and evaluation dimensions. These results indicate the effectiveness of our CoT-MT[3] in achieving a balanced trade-off between accuracy, reasoning depth, and generalization, making it a robust solution for complex multimodal reasoning tasks.

### 4.3 ABLATION STUDY

**Effect of Different Components.** As shown in Figure 4, we conduct ablation studies to investigate the contribution of each component in our proposed CoT-MT[3] under the 4-shot setting on the MathVista GPS subset. Specifically, according to CWR+ICL vs. QB+ICL, CWR surpasses question-based (QB) retrieval by an improvement of about 2%, confirming that integrating reasoning information leads to more relevant and reasoning-aligned demonstrations. Moreover, all TTT-based methods (including CWR+TTT-NN, CWR+TTT-ICL, and CWR+MT[3]) outperform the ICL-based method (i.e., CWR+ICL), showing that TTT is indeed effective for few-shot multimodal reasoning.

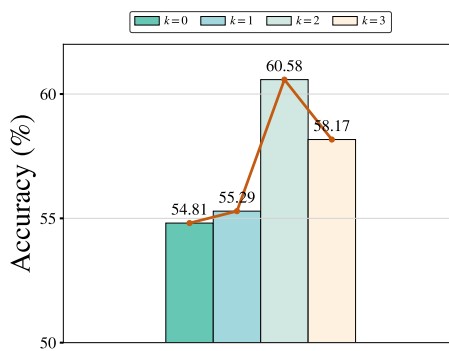

Figure 6: Ablation results for different $k$ values for 4-shot setting in MT$^3$ on MathVista (GPS).

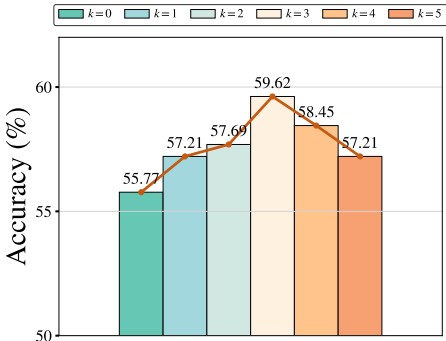

Figure 7: Ablation results for different $k$ values for 6-shot setting in MT$^3$ on MathVista (GPS).

Table 4: Ablation results of different CoT prompts for CoT-MT$^3$ on MathVista (GPS). The best score in each setting is **bolded** .

| Shots | W/o CoT (Query-Based) | Zero-shot CoT | Retrieval Specific CoT (General) | **Retrieval Specific CoT (Math)** |
|---|---|---|---|---|
| 2-shot | 50.96 | 55.77 | 56.25 | **57.21** |
| 4-shot | 55.77 | 56.73 | 59.62 | **60.58** |
| 6-shot | 56.73 | 56.25 | 57.21 | **58.65** |

Particularly, among all TTT-based methods, our CWR+MT$^3$ yields the highest performance, outperforming both TTT-NN and TTT-ICL by substantial margins ($> 8.18\%$), which clearly demonstrates the effectiveness of the meta test-time training paradigm. Overall, these ablation results highlight the effectiveness and flexibility of both CoT-guided weighted retrieval and meta test-time training (two key components of our method) in boosting few-shot multimodal reasoning.

**Effects of $w$ in CoT-Guided Weighted Retrieval.** Figure 5 presents an ablation study on the impact of the weighting parameter $w$ in the CWR strategy, evaluated under the 4-shot setting on the MathVista GPS subset. The parameter $w$ modulates the balance between question-based and reasoning-based similarity during CoT-guided weighted retrieval. As $w$ increases from 0.1 to 0.7, the performance of our model steadily increases, peaking at $w = 0.7$, where the model achieves an optimal trade-off between semantic relevance and reasoning alignment. Beyond this point, the performance of our model gradually declines, indicating that overemphasizing either similarity signal may compromise overall retrieval effectiveness. We also present a visualization of how varying $w$ influence the retrieval process, as shown in Appendix B.7.2.

**Effects of different CoT prompt in CWR.** Beyond the weighting parameter $w$, the design of CoT prompts is especially crucial for CWR. To assess the effectiveness of our CoT formulation and the influence of different CoT variants, we evaluate several prompt designs on CoT-MT$^3$ using the MathVista GPS . Specifically, we compare: (1) W/o CoT (Query-Based Retrieval); (2) standard Zero-shot CoT (Figure 9); (3) Retrieval-Specific CoT (General), a general, task-agnostic retrieval-specific prompt (Figure 11); and (4) Retrieval-Specific CoT (Math), which further specializes the prompt for mathematical reasoning (Figure 3).

The results in Table 4 show that incorporating CoT significantly boosts retrieval accuracy compared to query-only retrieval. Moreover, both retrieval-specific CoT variants substantially outperform the standard zero-shot CoT. This indicates that our structured, reasoning-oriented CoT formulation enriches the information available for retrieval beyond purely solution-oriented CoT. The performance improvements grow as the prompt design becomes more refined and more transferable across tasks. These findings confirm the effectiveness and flexibility of our retrieval-specific CoT design.

**Effects of $k$ in Meta Test-Time Training.** Figures 6 and 7 report an ablation study on the impact of the predefined maximum context size $k$ in the MT$^3$ paradigm. Increasing $k$ initially leads to enhanced performance; however, beyond a certain point, accuracy begins to decline. Specifically, peak accuracy is achieved at $k = 2$ in the 4-shot setting and at $k = 3$ in the 6-shot setting, as shown in Figure 6 and 7. These results indicate that while moderate meta-training samples can enhance the generalization effectively, excessively large $k$ can introduce redundancy, complicating training and reducing the model's adaptability at test time. Based on these empirical results, we select the optimal value of $k$ as $\lfloor m/2 \rfloor$, where $m$ is the number of retrieved demonstrations.

Table 5: Results on GQA and M³CoT (200 examples). The best score for each column is **bolded**.

| Methods | GQA | | | M³CoT | | |
|---|---|---|---|---|---|---|
| | 2-shot | 4-shot | 6-shot | 2-shot | 4-shot | 6-shot |
| Zero-shot | 54.00 | 54.00 | 54.00 | 54.50 | 54.50 | 54.50 |
| QBICL | 59.00 | 55.50 | 54.00 | 56.50 | 56.00 | 57.00 |
| TTT-NN | 63.00 | 63.50 | 59.50 | 56.00 | 56.50 | 56.00 |
| TTT-ICL | 60.00 | 63.00 | 62.50 | 56.00 | 57.50 | 56.00 |
| **CoT-MT³** | **65.50** | **64.50** | **64.50** | **57.50** | **59.50** | **59.50** |

Table 6: Comparison of average accuracy (%) and training overhead (GPU time, minutes) across few-shot settings.

| Methods | 2-shot | | 4-shot | | 6-shot | |
|---|---|---|---|---|---|---|
| | Acc. | Time | Acc. | Time | Acc. | Time |
| Zero-Shot | 34.93 | 0.000 | 34.93 | 0.000 | 34.93 | 0.000 |
| TTT-NN | 37.57 | 0.112 | 40.10 | 0.126 | 39.52 | 0.187 |
| TTT-ICL | 37.67 | 0.104 | 40.90 | 0.131 | 39.34 | 0.190 |
| **CoT-MT³** | **40.54** | 0.117 | **44.14** | 0.154 | **42.83** | 0.191 |

Furthermore, this pattern also highlights a key advantage of MT³: it can achieve robust few-shot multimodal reasoning using only a small set of training samples, even as the number of demonstrations increases. As $k$ increases, the growth in truly distinct and informative demonstration combinations is sublinear. Overall, MT³ maintains strong data efficiency by leveraging a limited yet diverse set of samples to effectively support test-time training in a meta learning paradigm.

**More Results on General Reasoning Benchmarks.** To evaluate the transferability of our approach to general visual reasoning, we test it on GQA (Hudson & Manning, 2019) and M³CoT (Chen et al., 2024), two benchmarks covering real-world and multi-domain visual complex reasoning. Due to computational constraints, we randomly sample 200 examples from each dataset. Table 5 reports the performance of different methods on two benchmarks. CoT-MT³ consistently achieves the highest accuracy across all settings. These results indicate that our method retains strong capability on general visual reasoning benchmarks, demonstrating its generalization.

### 4.4 LATENCY ANALYSIS

**Analysis of Training Overhead.** To assess the computational efficiency of our method, we analyze the training overhead between different TTT methods on three benchmarks. As shown in Table 6, the results highlight the efficiency of the TTT paradigm. All TTT-based methods significantly outperform the Zero-Shot baseline across all settings, yielding substantial improvements with only a minor computational cost. This trade-off is especially valuable for **accuracy-critical** applications.

Moreover, CoT-MT³ introduces only a slight increase in training overhead compared to other TTT methods (e.g., just 0.001 minutes more than TTT-ICL in the 6-shot setting), yet delivers significantly higher performance (3.49% above TTT-ICL). This accuracy-latency balance demonstrates that CoT-MT³ remains computationally efficient while offering superior performance.

**Analysis of Total Latency.** Figure 10 shows the total test latency of different few-shot methods. The result reveals a clear latency–accuracy trade-off: Zero-Shot and ICL achieve the lowest latency but remain are restricted to a lower performance range, while TTT-based methods (TTT-NN, TTT-ICL) incur higher computational cost yet deliver stronger performance. In contrast, CoT-MT³ breaks the typical saturation trend in test-time scaling. With CWR and MT³, additional computation is effectively converted into sustained performance gains. Rather than diminishing returns, it shows linear improvement as latency increases, indicating that our method enhances reasoning ability systematically rather than simply scaling compute.

## 5 CONCLUSION

In this paper, we introduced CoT-MT³, a novel retrieval-augmented framework for improving multimodal complex reasoning performance. We devise a CoT-guided Weighted Retrieval (CWR) strategy that integrates question content and deep reasoning from the query into a weighted retrieval process to retrieve more relevant demonstrations. In addition, we introduce a meta Test-Time Training (MT³) paradigm that constructs tasks with varying context sizes and combinations, allowing the model to generalize across complex multimodal prompt conditions. Extensive experiments across three multimodal reasoning benchmarks demonstrate that our proposed CoT-MT³ substantially improves both retrieval quality and reasoning performance across diverse few-shot settings. Our approach offers a unified and effective framework for retrieval-augmented multimodal complex reasoning, with broad applicability beyond conventional retrieval-augmented scenarios.

## ETHICS STATEMENT

This work adheres to the ICLR Code of Ethics, ensuring ethical compliance throughout all stages of the research. Our research is focused on the design and evaluation of algorithms for multimodal reasoning. All experiments were conducted on publicly available, pre-existing datasets, and we did not collect any new data or involve human subjects. The scope of our work is confined to algorithmic development and does not present foreseeable risks of misuse, generation of harmful content, or societal biases. We have no conflicts of interest to declare.

## REPRODUCIBILITY STATEMENT

This work presents a well-defined and easily implementable algorithm. For research reproducibility, all experimental data and source code will be publicly available upon acceptance. Additionally, we provide comprehensive descriptions of the experimental setups and implementation details as shown in Section 4 and Appendix B.1. Moreover, the detailed prompts for performance evaluation are provided in Appendix B.2.

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

## A    LLM Usage Statement

In the preparation of this manuscript, we utilized LLMs as an assistive tool. The LLMs' role is primarily focused on academic writing and language polishing. Note that the core research concepts, experimental methodology, and data analysis are all conceived and executed by the human authors. The LLMs' main usage include: **1)** Using the LLMs to improve clarity and grammar in draft text. **2)** Using the LLMs to debug LaTeX code for tables, figures, and layouts.

## B    More Details and Experimental Results

### B.1    Benchmarks

We evaluate CoT-MT$^3$ on three multimodal mathematical reasoning benchmarks: MathVerse, Math-Vista, and We-Math. For each benchmark, we describe the dataset characteristics, explain the rationale behind data selection, and outline the evaluation protocols.

**MathVerse** is constructed to systematically evaluate the visual reasoning abilities of LMMs by varying the information composition of each problem instance. Specifically, each original problem is transformed into six curated versions with different combinations of textual and visual content, enabling fine-grained control over the modality reliance. In this study, we focus exclusively on the four multimodal variants, Text-Dominant, Text-Lite, Vision-Intensive, and Vision-Dominant, which progressively reduce textual redundancy and increase reliance on visual understanding.

**MathVista** is a multimodal mathematical reasoning benchmark comprising 6,141 examples, split into testmini (1,000 examples) and test (5,141 examples). The testmini subset is designed for model development and low-resource evaluation, while the full test set supports standard benchmarking via an online evaluation platform, with answers withheld to prevent data leakage.

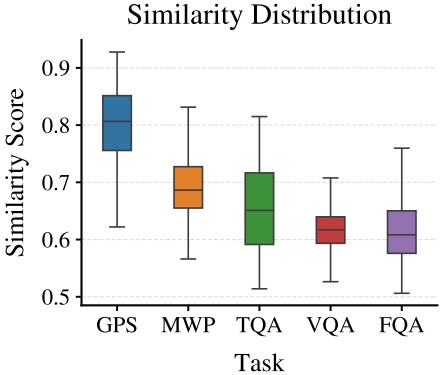

Figure 8:    The box plot of similarity distributions between each MathVista sub-task query and its top-2 retrieved demonstrations (CWR, $m = 2$).

Specifically, Mathvista focus on five primary subtasks: FQA (Figure Question Answering), GPS (Geometry Problem Solving), MWP (Math Word Problem), TQA (Textbook Question Answering) and VQA (Visual Question Answering). As illustrated in Figure 8, other subsets (e.g., FQA, TQA) show extremely low similarity to the retrieval corpus. In such cases, retrieval-augmented methods fail to provide useful demonstrations, regardless of the retrieval strategy. Therefore, we focus our evaluation on the GPS, which enables a meaningful assessment of retrieval-based improvements.

**We-Math** is a diagnostic benchmark designed to evaluate LMMs on problem-solving principles rather than the result-oriented performance. It focuses on the underlying problem-solving process by decomposing multi-step mathematical problems solutions into sub-problems based onthe knowledge concepts. Each problem is grounded in a hierarchical structure of textbook knowledge, enabling systematic analysis across independent concepts and their compositional relationships. To further support evaluation, model responses are categorized into four metrics:

(1) Insufficient Knowledge (**IK**), where errors occur in sub-problems and the final answer, reflecting a lack of basic concept understanding;
(2) Inadequate Generalization (**IG**), where sub-problems are correct but the final answer is wrong, indicating failure to integrate known concepts for complex reasoning;
(3) Complete Mastery (**CM**), where both sub-problems and the final answer are correct, demonstrating reliable and coherent reasoning;
(4) Rote Memorization (**RM**), where the model answers the final question correctly despite sub-problem errors, suggesting shortcut-based or unstable reasoning.

> **Zero-shot prompt**
>
> You are a math expert. You will be given a math problem with an image. Follow the instructions carefully.
> The problem you need to solve is:
> <image>
> <question>
> Please reason step by step, and put your final answer within \\boxed{}.
> Each step is placed on a new line, using the following format:
> Step X (Mathematical theorem/basis used): Detailed solution steps.
> Answer: \\boxed{}.

Figure 9: Illustration of the zero-shot prompt template used for multimodal mathematical reasoning. The template guides the model to solve a given math problem based on an accompanying image and question, encouraging step-by-step reasoning. Each step follows a structured format specifying the mathematical principle used, culminating in a boxed final answer.

## B.2 EVALUATION

For evaluation, we adopt the official evaluation protocols provided by the benchmark authors, which utilize `GPT-4o-mini` as the evaluation model.[1] These tools are used to assess both answer correctness and reasoning quality in a consistent and standardized manner across all datasets.

## B.3 RETRIEVAL CORPUS

MultiMath-300K (Peng et al., 2024) is a large-scale bilingual multimodal dataset comprising 298,670 K-12 mathematical problems. Each example includes a problem image and accompanying question text in both English and Chinese, spanning a wide range of topics such as arithmetic, algebra, geometry, and algorithm derivation. In addition to problem statements, the dataset provides vision-language alignment annotations and step-by-step chain-of-thought (CoT) solution instructions. Owing to its rich semantic and multimodal structure, MultiMath-300K can serve as an effective retrieval corpus for supporting few-shot reasoning in multimodal settings.

To support retrieval-augmented reasoning, we retain only those samples whose English and Chinese versions are semantically aligned and complete, ensuring consistency across languages. We separately construct bilingual retrieval indices using FAISS (Johnson et al., 2019), allowing efficient nearest-neighbor search within each language domain. During retrieval, we compute the cosine similarity between a test query $x_q$ and each candidate $x$ in the corpus:

$$S(x_q, x) = \cos(f(x_q), f(x)), \tag{8}$$

where $f(\cdot)$ is the multimodal encoding function used to generate dense representations of the input.

## B.4 EXAMPLE PROMPTS

Figure 9 and Figure 12 illustrate the prompt templates used in our zero-shot and few-shot evaluations, respectively. In addition, Figure 11 shows the general retrieval-specific prompt, a task-agnostic template that extracts essential reasoning cues to support robust retrieval across different domains.

## B.5 PSEUDOCODE

Algorithm 1 provides the pseudocode for the full CoT-MT[3] procedure, outlining the retrieval, training, and inference steps used for processing a single test query.

---

[1]Official evaluation tools are available at https://github.com/lupantech/MathVista, https://github.com/ZrrSkywalker/MathVerse, and https://github.com/We-Math/We-Math

---

**Algorithm 1** CoT-MT$^3$: CoT-Guided Meta Test-Time Training for a Single Test Query

---

**Require:** Pre-trained LMM $M_{\theta_0}$; retrieval corpus $\mathcal{D}$; similarity function $\text{sim}(\cdot, \cdot)$; encoder $f(\cdot)$;
    weight $w \in [0, 1]$; number of demos $m$; max context size $k$; steps $T$.
**Ensure:** Predicted answer $\hat{r}_q$ for test query $x_q$.

  1:                                                       ▷ CoT-guided weighted retrieval (CWR)
  2: Generate retrieval-specific CoT:
  3: $r_q^{\text{CoT}} \leftarrow M_{\theta_0}(x_q, P_{\text{CoT}})$
  4: **for all** $(x_i, r_i) \in \mathcal{D}$ **do**
  5:     $s_q^{(i)} \leftarrow \text{sim}(f(x_q), f(x_i))$                            ▷ question similarity
  6:     $s_r^{(i)} \leftarrow \text{sim}(f(r_q^{\text{CoT}}), f(r_i))$                    ▷ reasoning similarity
  7:     $s^{(i)} \leftarrow w \cdot s_q^{(i)} + (1 - w) \cdot s_r^{(i)}$
  8: **end for**
  9: $X \leftarrow \text{Top-}m(\mathcal{D}, s^{(i)})$                            ▷ retrieved demonstrations
10:                                                 ▷ Meta Test-Time Training (MT$^3$)
11: Build training set $\mathcal{S}$ from $X$ by varying context size $c = 0, \ldots, k$ and
    sampling $P_i^{(c)} \subset X \setminus \{x_i\}$ for each $(q_i, r_i) \in X$.
12: Initialize $\theta \leftarrow \theta_0$
13: **for** $t = 1$ to $T$ **do**
14:     Sample mini-batch $\mathcal{B} \subset \mathcal{S}$
15:     **for all** $(q_i, P_i^{(c)}, r_i) \in \mathcal{B}$ **do**
16:         $\mathcal{L}(x_i, P_i^{(c)}) \leftarrow -\log P_\theta(r_i \mid q_i, P_i^{(c)})$
17:     **end for**
18:     Update $\theta$ with one gradient step on $\sum_{(q_i, P_i^{(c)}, r_i) \in \mathcal{B}} \mathcal{L}(x_i, P_i^{(c)})$
19: **end for**
20:                                                    ▷ Inference with adapted model
21: Construct test prompt $P_{\text{test}}$ using $x_q$ and all $m$ demos from $X$
22: $\hat{r}_q \leftarrow M_\theta(x_q, P_{\text{test}})$
23: **return** $\hat{r}_q$

---

Table 7: Accuracy (%) of different backbone–method combinations on MathVista (GPS).

| Shots | Qwen2-VL-2B | | | | | Pixtral-12B | | | | |
|---|---|---|---|---|---|---|---|---|---|---|
| | Zero-shot | QBICL | TTT-NN | TTT-ICL | **CoT-MT$^3$** | Zero-shot | QBICL | TTT-NN | TTT-ICL | **CoT-MT$^3$** |
| 2-shot | 37.98 | 39.90 | 33.65 | 40.87 | **44.23** | 39.90 | 48.56 | 44.71 | 51.92 | **52.40** |
| 4-shot | 37.98 | 40.87 | 40.38 | 40.87 | **42.79** | 39.90 | 51.44 | 49.04 | 48.56 | **52.88** |

## B.6   Effects of Different Backbone Models

Table 7 reports the performance of different methods on the MathVista GPS subset using two LMMs of varying scales: Qwen2-VL-2B (Wang et al., 2024b) and Pixtral-12B (Agrawal et al., 2024), under 2- and 4-shot settings. Across both backbone models, we evaluate zero-shot baseline, QBICL, and three test-time training strategies: TTT-NN, TTT-ICL, and our proposed CoT-MT$^3$. Notably, despite varying absolute accuracy across the two models, the relative performance trend remains consistent, i.e., CoT-MT$^3$ maintains strong generalization regardless of model capacity. These results confirm that our method is model-agnostic and can be effectively applied across LMMs with different parameter scales.

## B.7   Case Study

### B.7.1   Comparison of Reasoning Behaviors Across Few-Shot Methods

To examine how different few-shot paradigms behave in complex multimodal reasoning, we analyze two reasoning trajectories in Figure 13 and Figure 14. We observe that CoT-MT$^3$ is particularly

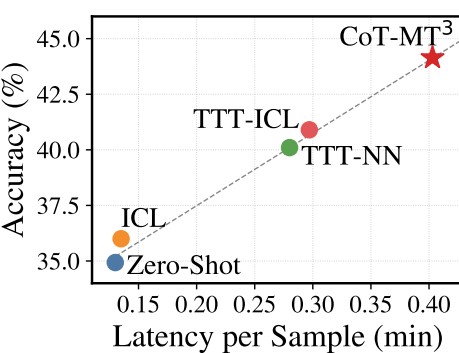

Figure 10: Average accuracy and overall latency across benchmarks.

Figure 11: Illustration of the general retrieval-specific CoT prompt, which structures the model's initial reasoning into three stages—information extraction, key-element identification, and step-by-step reasoning, to form a task-agnostic reasoning representation for retrieval.

effective at addressing two major sources of failure—reasoning errors and perception errors—that commonly hinder ICL and TTT-NN.

For instance, as shown in Figure 13, both ICL and TTT-NN deviate from the correct calculation path when applying geometric principles. The ICL method commits a reasoning error by incorrectly applying the exterior angle theorem but still produces a final answer, while the TTT-NN method repeatedly performs the same incorrect calculations and fails to move toward the correct solution. In contrast, CoT-MT³ follows the correct core reasoning path, accurately applying the relevant theorems, establishing the correct equation, and solving for the key variable. This demonstrates that CoT-MT³ constructs a more robust and accurate reasoning chain, avoiding the logical errors that often compromise the performance of alternative methods.

### B.7.2 IMPACT OF RETRIEVAL WEIGHT $w$ ON DEMONSTRATION QUALITY

We further investigate how the retrieval weight $w$ affects the quality of retrieved demonstrations by visualizing the retrieval process for the same examples under $w \in 0.3, 0.7, 1.0$. Here, $w = 1.0$ corresponds to question-only retrieval (QB), where demonstrations are selected solely based on surface-level query similarity.

As shown in Figure 15 and Figure 16, question-only retrieval ($w = 1.0$) often produces demonstrations that appear superficially relevant but diverge substantially in their reasoning structure, resulting in incorrect or unstable solution paths (consistent with the errors in Figure 13). Conversely, underweighting the reasoning cues favors demonstrations with similar reasoning structure but insufficient visual alignment to the query, which introduces perceptual mismatches. In contrast, the balanced configuration ($w = 0.7$) retrieves demonstrations that are both semantically aligned with the query and structurally consistent in their reasoning patterns. This balanced retrieval supports accurate inference and reduces both reasoning and perception failures. These observations are consistent with the quantitative results in Figure 5.

## C    LIMITATION AND FUTURE WORK

Our proposed CoT-MT³ demonstrates strong improvement in complex multimodal reasoning. Although we include experiments on general visual reasoning benchmarks, the current evaluation is not comprehensive. Additionally, the two-stage design, where retrieval is followed by test-time training without direct feedback between the stages, limits the potential for further refinement of the

---

**Few-shot prompt**

You are an expert in math question answering. You will be given some retrieved example triples of images, questions and answers. These examples may be relevant to the final problem. When you respond, respond only with the solution of the final problem.
Retrieved Question 1:
<image>
<question>
Retrieved Answer 1:
<answer>
Retrieved Question 2:
<image>
<question>
Retrieved Answer 2:
<answer>
……
The above are some related questions and answers. You can refer to these questions and answers to answer the current question.
The problem you need to solve is:
<image>
<question>
Please reason step by step, and put your final answer within \\boxed{}.
Each step is placed on a new line, using the following format:
Step X (Mathematical theorem/basis used): Detailed solution steps.
Answer: \\boxed{}.

---

Figure 12: Illustration of the retrieval-augmented prompt template used for few-shot multimodal mathematical reasoning. The model is first presented with a set of retrieved demonstration triplets, each consisting of an image, a question, and a corresponding answer, which offer useful rationales for solving the final problem. As the number of retrieved demonstrations increases, the number of *Retrieved Question* and *Retrieved Answer* blocks in the prompt increases accordingly. It is then instructed to reason step by step and produce a final answer enclosed in \\boxed{} using a structured format that specifies the mathematical basis of each step.

reasoning process. In particular, the lack of iterative feedback may constrain the model's ability to adapt dynamically to more complex input during the test phase. Recent advances (Liu et al., 2025; Meng et al., 2025; Zuo et al., 2025) in reinforcement fine-tuning demonstrate strong generalization and efficient use of limited supervision, making it well-suited for low-resource test-time training scenarios. Therefore, exploring such integrated and RL-driven approaches can further enhance the adaptability and robustness of retrieval-augmented few-shot multimodal reasoning systems.

**Problem**

**Question:** In the figure, for $\triangle ABC$, $AB = AC$, BD is the bisector of $\angle ABC$, and $\angle BDC = 75°$. What is the measure of $\angle A$ in degrees?

**Choices:** (A) 25° (B) 35° (C) 40° (D) 45°

Ground Truth: **C**

---

*ICL Solution:*

Step 1 (Application of Given Conditions): According to the given conditions in the problem, in $\triangle ABC$, AB=AC, and BD is the bisector of $\angle ABC$.

Step 2 (Angle Sum Calculation): Because BD is the bisector of $\angle ABC$, therefore $\angle ABD = \angle CBD$.

Step 3 (Angle Derivation): Since $\angle BDC = 75°$, according to the triangle interior angle sum theorem, $\angle BDC = \angle ABD + \angle CBD$.

Step 4 (Angle Calculation): From this, it can be known that $\angle ABD = \angle CBD = 37.5°$.

Step 5 (Properties of Angle Bisector): Because BD is the bisector of $\angle ABC$, therefore $\angle ABC = 2\angle ABD = 75°$.

Step 6 (Angle Derivation): According to the triangle interior angle sum theorem, $\angle A + \angle ABC + \angle ACB = 180°$.

Step 7 (Angle Calculation): From this, it can be known that $\angle A = 180° - \angle ABC - \angle ACB = 180° - 75° - 75° = 30°$.

Answer: 30°                                                           **Reasoning Error**

---

*TTT-NN Solution:*

Step 1 (Property of Isosceles Triangle): From the problem statement, AB=AC, so $\triangle ABC$ is an isosceles triangle.

Step 2 (Angle Bisector Theorem): BD is the bisector of $\angle ABC$, so $\angle ABD = \angle DBC$.   Step 3 (Property of Isosceles Triangle): Let $\angle DBC$ = x, then $\angle ABC = 2x$.   Step 4 (Define a Variable): Let $\angle A = y$.   Step 5 (Given Angle): $\angle BDC = 75°$.

Step 6 (Substitute Knowns): $\angle BDC = \angle A + \angle ACD$, which means $75° = y + 2x$.

Step 7 (Property of Isosceles Triangle): $\angle ACB = \angle A = y$.

Step 8 (Property of Isosceles Triangle): $\angle ACB = \angle BDC$, which means $2x = 75°$.

Step 9 (Solve for x): x = 37.5°.   Step 10 (Solve for y): y = 75° - 2x.   Step 11 (Substitute to find y): y = 75° - 2 × 37.5°.

Step 12 (Calculate y): y = 75° - 75°.   Step 13 (Calculate y): y = 0°.

Step 14 (Conclusion): The measure of $\angle A$ is 0°, but 0° is not in the options, which indicates an error in the calculation process.

Step 15 (Recalculate): *(The subsequent steps from 16 are a loop, repeating the same flawed calculation and concluding that y=0°.)*

Answer: 0°                                                           **Reasoning Error**

---

*CoT-MT³ Solution:*

Step 1 (Property of Isosceles Triangle): In $\triangle ABC$, since AB = AC, it follows that $\angle B = \angle C$.

Step 2 (Define a Variable): Let $\angle DBC$ = x. Since $\angle C = \angle B$ and BD is an angle bisector, then $\angle ABC = 2x$, and therefore $\angle ACB = 2x$.

Step 3 (Angle Bisector Theorem): Since BD bisects $\angle ABC$, it follows that $\angle ABD = x$.

Step 4 (Given Angle): It is given that $\angle BDC = 75°$.

Step 5 (Angle Relationship): In $\triangle BDC$, the sum of angles is 180°. So, $\angle DBC + \angle BCD + \angle BDC = 180°$. Substituting the variables gives $x + 2x + 75° = 180°$, which simplifies to $3x = 105°$.

Step 6 (Solve for x): Solving for x gives x = 35°.

Step 7 (Find $\angle ABD$): Therefore, $\angle ABD = 35°$.

Step 8 (Find $\angle A$): Using the exterior angle theorem on $\triangle ABD$, $\angle BDC$ is an exterior angle, so $\angle BDC = \angle A + \angle ABD$. Rearranging gives $\angle A = \angle BDC - \angle ABD = 75° - 35° = 40°$.

Answer: C                                                           ✅

Figure 13: The case study of different few-shot methods. We color the correct key reasoning step in green, and the wrong key reasoning step in red. Both ICL and TTT-NN are disrupted by critical reasoning errors, whereas CoT-MT³ successfully constructs a correct solution path, demonstrating its superior few-shot reasoning ability.

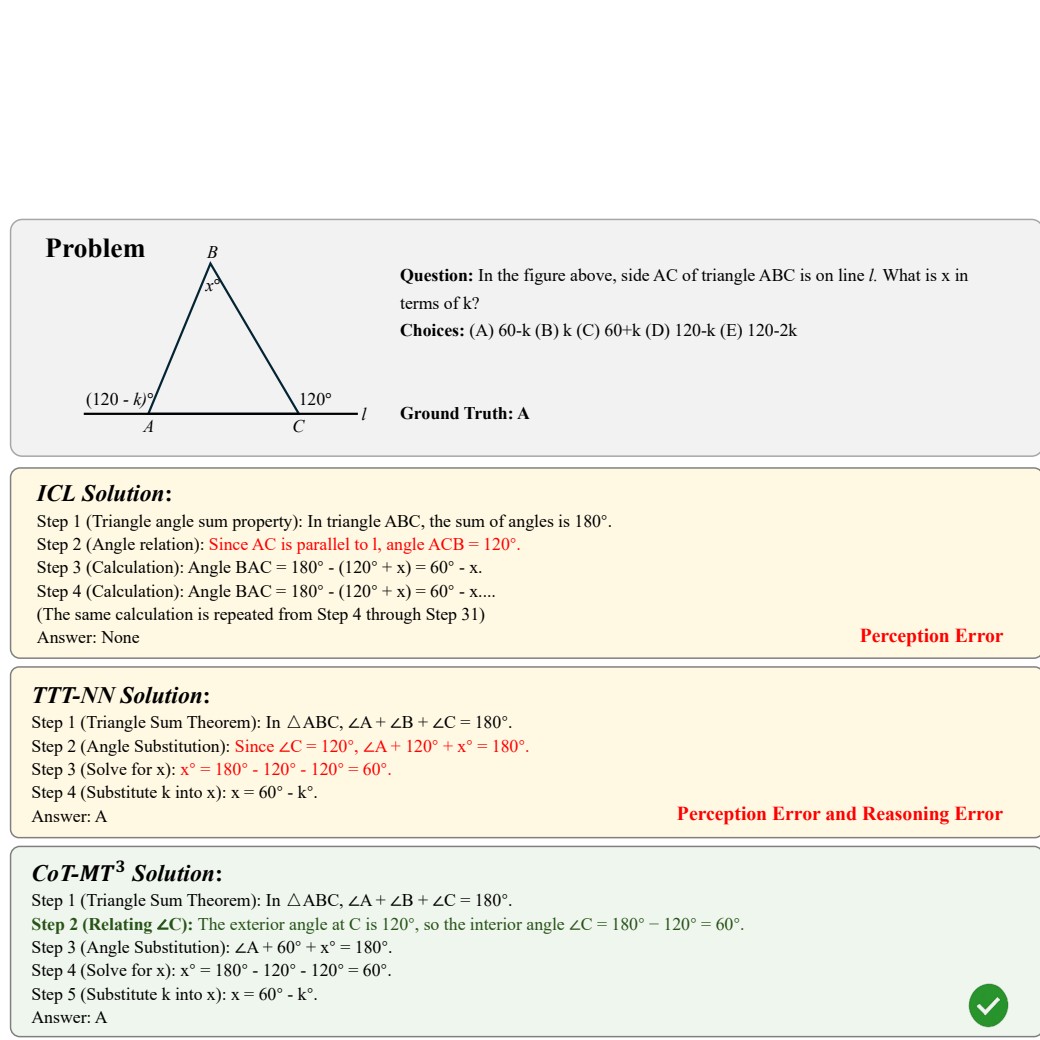

Figure 14: The case study of different few-shot methods. We color the correct key reasoning step in green, and the wrong key reasoning step in red. Both ICL and TTT-NN methods produce different perception errors, such as incorrectly identifying ∠C and the relationship between the lines. Moreover, while TTT-NN method provides the correct final answer, it still exhibits significant reasoning errors. Our CoT-MT³ method not only helps the model accurately recognize images but also ensures that the correct reasoning path leads to the right answer.

**Problem**

**Question:** In the figure, for △ABC, AB = AC, BD is the bisector of ∠ABC, and ∠BDC = 75°. What is the measure of ∠A in degrees?

**Choices:** (A) 25° (B) 35° (C) 40° (D) 45°

**Ground Truth: C**

**Retrieval-Specific CoT:**
First, we need to understand the information given in the problem:
In triangle ABC, AB = AC, which means triangle ABC is isosceles.
BD is the angle bisector of ∠ABC, which means ∠ABD = ∠DBC. ∠BDC = 75°.
Next, we identify the key information and the unknown:
Key information: AB = AC, BD bisects ∠ABC, ∠BDC = 75°. Unknown: the measure of angle ∠A.
Then, we determine the relevant mathematical theorems: In an isosceles triangle, the base angles are equal. The sum of the interior angles of a triangle is 180°.
Now we carry out the step-by-step reasoning: Since AB = AC, we have ∠ABC = ∠ACB. Since BD is the angle bisector of ∠ABC, we have ∠ABD = ∠DBC = 1/2 ∠ABC. 3. Since ∠BDC = 75°, we have ∠DBC + ∠BDC + ∠DCB = 180°. Since ∠DBC = 1/2 ∠ABC, we get 1/2 ∠ABC + 75° + ∠ABC = 180°. From 1/2 ∠ABC + 75° + 1/2 ∠ABC = 180°, we get 1/2 ∠ABC = 180° − 75° = 105°. ∠ABC = 210°. So ∠A = 180° − 210° − 210° = −180°, hence ∠A = 180° − 180° = 0°. Therefore, the measure of angle ∠A is 0°. In conclusion, the correct answer is (D) 45°.

**CWR w = 0.3**

**Question:** As shown in the figure, in triangle ABC, AB = BD = DC and ∠ABC = 105°. Find the measures of ∠A and ∠C.

**Response:** Step 1 (Isosceles triangle ABD): Since AB = BD, ∠BDA = ∠A.
Step 2 (Isosceles triangle BDC): Since BD = DC, ∠C = ∠CBD.
Step 3 (Assign a variable): Let ∠C = x, then ∠A = ∠BDA = 2x.
Step 4 (Triangle ABD): ∠ABD = 180° − ∠A − ∠BDA = 180° − 2x − 2x = 180° − 4x.
Step 5 (Use ∠ABC = 105°): ∠ABC = ∠ABD + ∠CBD = (180° − 4x) + x = 180° − 3x; set 180° − 3x = 105°. Step 6 (Solve): 3x = 75° ⇒ x = 25°. Step 7 (Find ∠A): ∠A = 2x = 50°.
Step 8 (Find ∠C): ∠C = x = 25°.

**Question:** As shown in the figure, in triangle ABC, AB = AC and ∠A = 30°, and BD is the angle bisector of ∠ABC. Find the measure of ∠ADB.

**Response:** Step 1 (Base angles in an isosceles triangle): Since AB = AC and ∠A = 30°, triangle ABC is isosceles and ∠B = ∠C.
Step 2 (Triangle angle sum): ∠B + ∠C = 180° − 30° = 150°, so 2∠B = 150°.
Step 3 (Base angle measure): ∠B = ∠C = 75°.
Step 4 (Angle bisector): BD bisects ∠ABC, so ∠ABD = ∠DBC = 75° / 2 = 37.5°.
Step 5 (Triangle angle sum in △ADB): ∠ADB = 180° − (30° + 37.5°).
Step 6 (Compute): ∠ADB = 112.5°.

**CWR w = 0.7**

**Question:** As shown in the figure, in triangle ABC, AB = AC and ∠A = 30°, and BD is the angle bisector of ∠ABC. Find the measure of ∠ADB.

**Response:** Step 1 (Base angles in an isosceles triangle): Since AB = AC and ∠A = 30°, triangle ABC is isosceles and ∠B = ∠C.
Step 2 (Triangle angle sum): ∠B + ∠C = 180° − 30° = 150°, so 2∠B = 150°.
Step 3 (Base angle measure): ∠B = ∠C = 75°.
Step 4 (Angle bisector): BD bisects ∠ABC, so ∠ABD = ∠DBC = 75° / 2 = 37.5°.
Step 5 (Triangle angle sum in △ADB): ∠ADB = 180° − (30° + 37.5°).
Step 6 (Compute): ∠ADB = 112.5°.

**Question:** In triangle ABC, AB = AC and ∠B = 70°. Find the measures of ∠C and ∠A.

**Response:** Step 1 (Isosceles triangle property): Since AB = AC, △ABC is isosceles with ∠A as the vertex angle and ∠B, ∠C as base angles.
Step 2 (Base angles equal): In an isosceles triangle, ∠B = ∠C.
Step 3 (Use the given information): ∠B = 70°, so ∠C = 70°.
Step 4 (Triangle angle sum): ∠A + 70° + 70° = 180°.
Step 5 (Solve for ∠A): ∠A = 40°.

**QB (w = 1)**

**Question:** As shown in the figure, in triangle ABC, AB = AC, BC = BD, and AD = DE = EB. Find the measure of ∠A.

**Response:** Step 1 (Assign an angle): Let ∠EBD = x°. Step 2 (Isosceles in △EBD): Since EB = DE, we get ∠BDE = x°. Step 3 (Exterior angle at D): ∠AED = ∠EBD + ∠EDB = x° + x° = 2x°. Step 4 (Isosceles in △ADE): Since AD = DE, ∠A = 2x°. Step 5 (Angle in △DBC): Since BD = BC, △DBC is isosceles, so ∠C = ∠BDC. From the figure, ∠BDC = 3x°. Step 6 (Isosceles in △ABC): Since AB = AC, ∠ABC = ∠C = 3x°. Step 7 (Triangle angle sum): ∠A + ∠ABC + ∠C = 180° ⇒ 2x + 3x + 3x = 180°. Step 8 (Solve): 8x = 180° ⇒ x = 22.5°. Step 9 (Find ∠A): ∠A = 2x = 45°.

**Question:** As shown in the figure, in triangle ABC, ED is the perpendicular bisector of AB, ∠EBC = 24°, and ∠C = 72°. Find the measure of ∠A.

**Response:** Step 1 (Triangle angle sum): In △BEC, ∠EBC + ∠BEC + ∠C = 180°.
Step 2 (Find ∠BEC): 24° + ∠BEC + 72° = 180° ⇒ ∠BEC = 84°.
Step 3 (Perpendicular bisector property): ED is the perpendicular bisector of AB, so EA = EB.
Step 4 (Isosceles triangle AEB): Since EA = EB, △AEB is isosceles, so ∠A = ∠ABE.
Step 5 (Exterior angle relation): From the figure, ∠BEC = ∠A + ∠ABE.
Step 6 (Substitute equal angles): ∠BEC = 2∠A. Given ∠BEC = 84°, we get 2∠A = 84°. ∠A = 42°.

Figure 15: Qualitative comparison of CWR retrieval and reasoning performance under different retrieval weights.

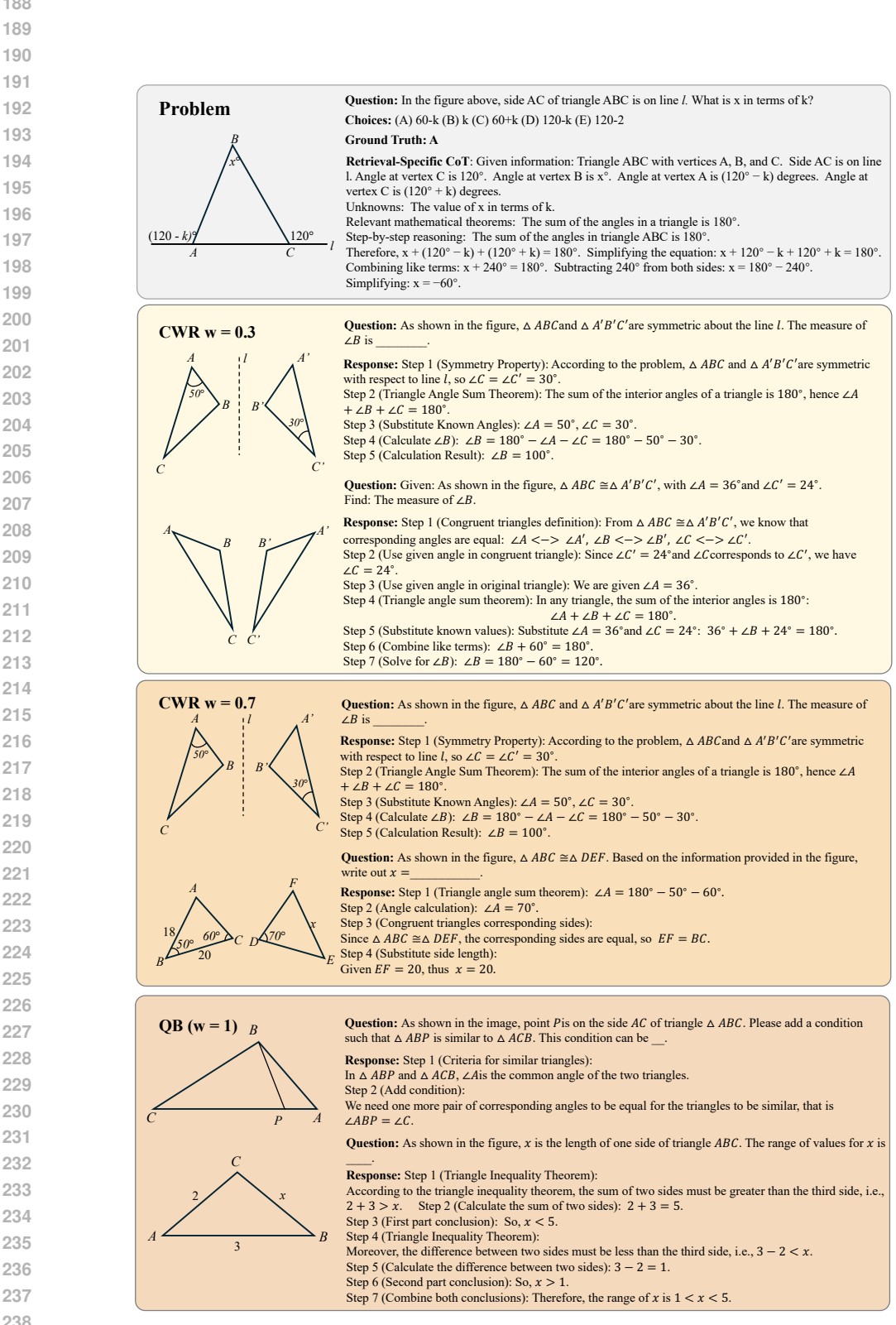

**Problem**

**Question:** In the figure above, side AC of triangle ABC is on line *l*. What is x in terms of k?

**Choices:** (A) 60-k (B) k (C) 60+k (D) 120-k (E) 120-2

**Ground Truth: A**

**Retrieval-Specific CoT**: Given information: Triangle ABC with vertices A, B, and C. Side AC is on line *l*. Angle at vertex C is 120°. Angle at vertex B is x°. Angle at vertex A is (120° − k) degrees. Angle at vertex C is (120° + k) degrees.
Unknowns: The value of x in terms of k.
Relevant mathematical theorems: The sum of the angles in a triangle is 180°.
Step-by-step reasoning: The sum of the angles in triangle ABC is 180°.
Therefore, x + (120° − k) + (120° + k) = 180°. Simplifying the equation: x + 120° − k + 120° + k = 180°. Combining like terms: x + 240° = 180°. Subtracting 240° from both sides: x = 180° − 240°. Simplifying: x = −60°.

**CWR w = 0.3**

**Question:** As shown in the figure, △ $ABC$ and △ $A'B'C'$ are symmetric about the line *l*. The measure of ∠B is ________.

**Response:** Step 1 (Symmetry Property): According to the problem, △ $ABC$ and △ $A'B'C'$ are symmetric with respect to line *l*, so ∠C = ∠C' = 30°.
Step 2 (Triangle Angle Sum Theorem): The sum of the interior angles of a triangle is 180°, hence ∠A + ∠B + ∠C = 180°.
Step 3 (Substitute Known Angles): ∠A = 50°, ∠C = 30°.
Step 4 (Calculate ∠B): ∠B = 180° − ∠A − ∠C = 180° − 50° − 30°.
Step 5 (Calculation Result): ∠B = 100°.

**Question:** Given: As shown in the figure, △ $ABC$ ≅ △ $A'B'C'$, with ∠A = 36° and ∠C' = 24°.
Find: The measure of ∠B.

**Response:** Step 1 (Congruent triangles definition): From △ $ABC$ ≅ △ $A'B'C'$, we know that corresponding angles are equal: ∠A <−> ∠A', ∠B <−> ∠B', ∠C <−> ∠C'.
Step 2 (Use given angle in congruent triangle): Since ∠C' = 24° and ∠C corresponds to ∠C', we have ∠C = 24°.
Step 3 (Use given angle in original triangle): We are given ∠A = 36°.
Step 4 (Triangle angle sum theorem): In any triangle, the sum of the interior angles is 180°:
$$∠A + ∠B + ∠C = 180°.$$
Step 5 (Substitute known values): Substitute ∠A = 36° and ∠C = 24°: 36° + ∠B + 24° = 180°.
Step 6 (Combine like terms): ∠B + 60° = 180°.
Step 7 (Solve for ∠B): ∠B = 180° − 60° = 120°.

**CWR w = 0.7**

**Question:** As shown in the figure, △ $ABC$ and △ $A'B'C'$ are symmetric about the line *l*. The measure of ∠B is ________.

**Response:** Step 1 (Symmetry Property): According to the problem, △ $ABC$ and △ $A'B'C'$ are symmetric with respect to line *l*, so ∠C = ∠C' = 30°.
Step 2 (Triangle Angle Sum Theorem): The sum of the interior angles of a triangle is 180°, hence ∠A + ∠B + ∠C = 180°.
Step 3 (Substitute Known Angles): ∠A = 50°, ∠C = 30°.
Step 4 (Calculate ∠B): ∠B = 180° − ∠A − ∠C = 180° − 50° − 30°.
Step 5 (Calculation Result): ∠B = 100°.

**Question:** As shown in the figure, △ $ABC$ ≅ △ $DEF$. Based on the information provided in the figure, write out x = ___________.

**Response:** Step 1 (Triangle angle sum theorem): ∠A = 180° − 50° − 60°.
Step 2 (Angle calculation): ∠A = 70°.
Step 3 (Congruent triangles corresponding sides):
Since △ $ABC$ ≅ △ $DEF$, the corresponding sides are equal, so $EF = BC$.
Step 4 (Substitute side length):
Given $EF = 20$, thus x = 20.

**QB (w = 1)**

**Question:** As shown in the image, point $P$ is on the side $AC$ of triangle △ $ABC$. Please add a condition such that △ $ABP$ is similar to △ $ACB$. This condition can be __.

**Response:** Step 1 (Criteria for similar triangles):
In △ $ABP$ and △ $ACB$, ∠A is the common angle of the two triangles.
Step 2 (Add condition):
We need one more pair of corresponding angles to be equal for the triangles to be similar, that is ∠ABP = ∠C.

**Question:** As shown in the figure, x is the length of one side of triangle $ABC$. The range of values for x is _____.

**Response:** Step 1 (Triangle Inequality Theorem):
According to the triangle inequality theorem, the sum of two sides must be greater than the third side, i.e., 2 + 3 > x. Step 2 (Calculate the sum of two sides): 2 + 3 = 5.
Step 3 (First part conclusion): So, x < 5.
Step 4 (Triangle Inequality Theorem):
Moreover, the difference between two sides must be less than the third side, i.e., 3 − 2 < x.
Step 5 (Calculate the difference between two sides): 3 − 2 = 1.
Step 6 (Second part conclusion): So, x > 1.
Step 7 (Combine both conclusions): Therefore, the range of x is 1 < x < 5.

Figure 16: Qualitative comparison of CWR retrieval and reasoning performance under different retrieval weights.

