# OpenReview forum: "CoT-MT$^3$ : CoT-Guided Meta Test-Time Training for Multimodal Reasoning"
_ICLR.cc/2026/Conference — Submitted to ICLR 2026_

### Official Review · Reviewer_CNNH · 2025-10-15

**Soundness:** 3
**Presentation:** 3
**Contribution:** 2
**Rating:** 4
**Confidence:** 4

**Summary:**

This paper presents CoT-MT3, a retrieval-augmented test-time training framework for improving few-shot multimodal reasoning in Large Multimodal Models (LMMs). The method integrates two components: (1) CoT-guided Weighted Retrieval (CWR), which incorporates structured retrieval-specific chain-of-thought (CoT) reasoning into the retrieval process; and (2) Meta Test-Time Training (MT3), which introduces a context-based meta-learning paradigm that trains LMMs using combinations of few-shot demonstrations with varying context sizes. Experiments on three multimodal math reasoning benchmarks—MathVerse, MathVista, and We-Math—show consistent performance gains over existing baselines. The framework demonstrates robustness across multiple few-shot settings and is claimed to be generalizable across diverse reasoning scenarios.

**Strengths:**

The idea of generating a retrieval-specific Chain-of-Thought (CoT) to improve the quality of retrieved examples makes a lot of sense. It goes beyond just ``question similarity'' and helps the model focus on reasoning-relevant content during retrieval.

The method is evaluated on three different multimodal math reasoning benchmarks (MathVerse, MathVista, and We-Math), across 2/4/6-shot settings. The improvements over strong baselines like TTT-ICL and QBICL are consistent. While the gains are not massive in every case, they’re steady and meaningful.

The paper includes detailed ablation studies on key design choices—such as the weight parameter $w$ in retrieval, the context size $k$ in MT3, and comparisons among different TTT strategies. This shows the authors have carefully examined what parts of the method actually contribute.

Few-shot multimodal reasoning—especially under test-time training constraints—is still an open and under-researched area. This paper proposes a fairly practical and modular framework that addresses it directly, and can likely be applied to related settings with minor modifications.

**Weaknesses:**

The paper describes MT3 as a ``meta-learning paradigm,'' but it doesn’t really align with the standard definition of meta-learning. There is no inner/outer loop, no bi-level optimization, and no explicit task adaptation process. What the authors actually implement is more of a prompt diversification strategy, where the model is trained with varied context sizes. This is closer to robust context mixing rather than genuine meta-learning. The terminology could easily mislead readers expecting MAML-like or Reptile-style formulations.

The approach heavily depends on the quality of the retrieval-specific Chain-of-Thought (CoT) generated before retrieval. However, no analysis is provided for scenarios where the CoT is flawed or misleading. For instance, if the CoT incorrectly identifies relevant theorems or focuses on the wrong reasoning steps, how does that affect retrieval and final prediction? Some robustness evaluation or failure-case analysis would make the method more convincing.

While the CoT-guided weighted retrieval (CWR) is well-motivated, the paper provides minimal insight into how the weighted combination of question-based and reasoning-based similarity actually impacts retrieval outcomes. The ablation on the weight parameter $w$ is helpful but not sufficient; qualitative examples or analysis of diversity versus redundancy among retrieved demonstrations would add depth. Additionally, the retriever (Vista) is treated as a black box—its suitability for this specific task is not discussed.

CoT-MT3 performs multiple heavy operations at test time: CoT generation, retrieval from a large corpus, sampling of multiple prompt combinations, and fine-tuning with LoRA. Despite this, there is no mention of computational cost, latency, or memory footprint. Without runtime comparisons against baselines (e.g., ICL or TTT-NN), it is hard to judge the practical efficiency of the method.

Although MT3 aims to improve robustness by varying context sizes, it still performs gradient-based updates on a small number of retrieved examples. There is no evidence showing whether the model avoids overfitting or simply memorizes spurious patterns from the demonstrations. A comparison against simpler regularization strategies (e.g., dropout, prompt shuffling) could help clarify this.

Several important hyperparameters and architectural details are fixed without justification. For example, the LoRA configuration (rank 8, $\alpha{=}16$) is not ablated, and the similarity function $\text{sim}(f(x), f(y))$ is not formally defined—are image and text embeddings equally weighted, or are they learned separately? These omissions weaken reproducibility and interpretability.

All experiments are focused on multimodal mathematical reasoning tasks. Since the retrieval-specific CoT is explicitly math-oriented (extracting theorems, listing knowns, etc.), it is unclear whether the same method would work for general multimodal reasoning (e.g., VQA, diagram understanding). A small-scale test on a non-math domain would greatly strengthen the generalization claim.

The paper states that code will be released upon acceptance, but for a complex method combining retrieval, CoT generation, and fine-tuning, reproducibility is difficult without pseudo-code or algorithmic steps. Including a concise algorithm block (for CWR and MT3) would make the contribution clearer and easier to reproduce.

**Questions:**

You mention that a retrieval-specific CoT is produced for each test query, but it’s not fully clear whether this is generated by the same LMM used for final prediction, or a separate frozen module. Is it generated in a single pass? Is there any sampling or filtering involved? Also, how sensitive is the final performance to the quality of this CoT?

For both question-based and reasoning-based similarity, what exactly is the feature encoder $f(\cdot)$? Is it Vista, Qwen2-VL, or another model? Are the embeddings frozen or fine-tuned? Are text and image features handled jointly or separately?

Why fix LoRA rank to 8 and $\alpha = 16$? It would be useful to know whether these settings are optimal. Did you experiment with other ranks or values of $\alpha$? Would full fine-tuning yield similar gains? An ablation here would help understand where the gains are coming from.


Since your method involves CoT generation, retrieval, context sampling, and LoRA-based adaptation, how long does it take per test instance? Can you provide average latency or throughput numbers, compared to baselines like TTT-ICL or QBICL? How robust is CoT-MT3 when retrieval is poor?
If the top-$m$ retrieved demos are irrelevant or noisy, does MT3 still help? Or does it overfit to bad demonstrations? Did you try testing under retrieval noise or random demo injection?


The CoT prompting template is clearly math-specific (e.g., extract knowns, theorems, etc.). Would your method generalize to other multimodal reasoning domains like VQA, commonsense, or procedural understanding? What adaptations would be needed? How large is the retrieval index and how is it built? You mention using MultiMath-300K for retrieval—do you use the entire corpus? What kind of retriever is used (e.g., FAISS, Vista, dense transformer)? What’s the average retrieval time and memory usage?

Why not include pseudo-code or algorithm blocks? Figure 2 is helpful, but a pseudocode version of MT3 and CWR would really improve clarity and reproducibility. Any plans to add that in the appendix? Did you try disabling reasoning similarity (i.e., $w = 1.0$)? It would be helpful to know how much the reasoning part contributes to retrieval. A comparison between question-only ($w = 1.0$) and your mixed setup would make this more concrete.

\textit{If you’re able to address these questions with clarity and detail, especially around the retrieval/CoT generation process and efficiency, I’m open to increasing my score.}

---

> ### Author Response · Authors · 2025-11-24
> **Authors' Response (1/4)**
>
> We thank the reviewer for the thorough and insightful feedback. To avoid redundancy, we group related comments and address them jointly below, while explicitly referencing the corresponding IDs (W1–W8, Q1–Q6).
>
> **R1. On whether MT$^3$ is a meta-learning paradigm (W1)**
>
> **The paper describes MT$^3$ as a meta-learning paradigm, but it doesn’t really align with the standard definition of meta-learning. This is closer to robust context mixing rather than genuine meta-learning. The terminology could easily mislead readers expecting MAML-like or Reptile-style formulations. (W1)**
>
> **A:** Thank you for your insightful comment. The term "meta-learning" is often associated with approaches like MAML, which focus on learning a model that can quickly adapt to new unseen tasks. Our use of the term meta-learning follows a line of work in test-time training (TTT) and meta in-context learning [1]. Moreover, prior work [2] has shown that the number of demonstrations strongly influences task difficulty in reasoning tasks.
>
> Motivated by this, MT$^3$ constructs a set of training tasks by pairing each query with different context sizes and trains the model to remain accurate as the test context becomes longer and more complex. Through MT$^3$, the model can learn how to utilize context for effective reasoning during test time, which is precisely the meta-learning perspective we intend to convey. We will clarify this broader meta-learning interpretation in the revision version.
>
> [1] MetaICL: Learning to Learn In Context. NAACL 2022.
>
> [2] What Factors Affect Multi-Modal In-Context Learning? An In-Depth Exploration. NIPS 2024.
>
> **R2. On computational cost, latency (W4, Q4)**
>
> **How long does it take per test instance?  What is the average latency or throughput numbers, compared to baselines like TTT-ICL or QBICL? Without runtime comparisons against baselines (e.g., ICL or TTT-NN), it is hard to judge the practical efficiency of the method. (W4, Q4)**
>
> **A:** Thanks for pointing out the importance of latency and computational cost. We report a detailed analysis of latency and computational cost in Appendix B.5, and we will move a summary of these results into the main paper in the revised version.  All experiments are conducted on 4 NVIDIA A800-80G GPUs.
>
> Table 4 compares average accuracy and GPU time per test sample for different TTT methods. The table 4 is as follows:
>
> **Table: Training overhead of different TTT methods per query**
>
> | GPU time (mins)   | 2-shot | 4-shot | 6-shot |
> | ----------------- | ------ | ------ | ------ |
> | TTT-NN            | 0.1115 | 0.1258 | 0.1867 |
> | TTT-ICL           | 0.1044 | 0.1313 | 0.1892 |
> | **MT$^3$ (ours)** | 0.1168 | 0.1539 | 0.1913 |
>
> CoT-MT$^3$ introduces only a very small additional training overhead compared to existing TTT methods, while yielding higher accuracy. Additionally, Figure 10 plots the accuracy–latency trade-off for all methods. CoT-MT$^3$ lies on the best part of this curve, continuing to benefit from additional compute rather than saturating early. Given that our focus lies on **accuracy-critical** scenarios, especially for complex reasoning, we view a modest increase in per-query compute is acceptable in exchange for significantly improved multimodal reasoning.

---

> ### Author Response · Authors · 2025-11-24
> **Authors' Response (2/4)**
>
> **R3. on the retrieval/CoT generation process (W2, W3, Q1, Q4, Q6).**
>
> **The approach heavily depends on the quality of the retrieval-specific Chain-of-Thought (CoT) generated before retrieval. However, no analysis is provided for scenarios where the CoT is flawed or misleading. For instance, if the CoT incorrectly identifies relevant theorems or focuses on the wrong reasoning steps, how does that affect retrieval and final prediction? Some robustness evaluation or failure-case analysis would make the method more convincing. Did you try testing under retrieval noise or random demo injection? Did you try disabling reasoning similarity (i.e., )? It would be helpful to know how much the reasoning part contributes to retrieval. A comparison between question-only () and your mixed setup would make this more concrete. (W2, Q4, Q6)**
>
> **A:** Thanks for the helpful suggestions. In practice, due to the limited capability of the base model, its CoT is often noisy: as illustrated in Appendix B.8, the CoT for solving reasoning problems always include incorrect intermediate steps (e.g., the reasoning error in Figure 15) or mis-identified theorems. Our retrieval-specific CoT prompt is designed to mitigate this by decomposing the problem into multiple structured fields (e.g., key information, relevant knowledge, intermediate goals), providing a richer and more stable description than the raw question alone.
>
> To more concretely analyze the contribution and robustness of the reasoning-based retrieval signal, we conduct an ablation study on MathVista GPS, comparing:
> 1. **W/o CoT (Query-Based Retrieval):** disabling reasoning similarity;
> 2. **Zero-shot CoT:** using standard zero-shot CoT as the retrieval signa (Figure 9);
> 3. **Retrieval-Specific CoT (General):** using our general retrieval-specific CoT prompt (Figure 12);
> 4. **Retrieval-Specific CoT (Math):** using the math-tailored retrieval-specific CoT prompt (Figure 3).
>
> The results are as follows:
>
> **Table: Ablation of CoT prompt types for CoT-MT$^3$ on MathVista (GPS).**
>
> | CoT Prompt Type                  | 2-shot    | 4-shot    | 6-shot    |
> | -------------------------------- | --------- | --------- | --------- |
> | W/o CoT (Query-Based Retrieval)  | 50.96     | 55.77     | 56.73     |
> | Zero-shot CoT                    | 55.77     | 56.73     | 56.25     |
> | Retrieval-Specific CoT (General) | **57.21** | 59.13     | 57.21     |
> | Retrieval-Specific CoT (Math)    | **57.21** | **60.58** | **58.65** |
>
> Several observations address the reviewer’s concerns:
>
> - Comparing **W/o CoT** (question-only retrieval) with the two retrieval-specific CoT prompts shows that incorporating reasoning information substantially improves retrieval quality and final accuracy, directly quantifying the benefit of the reasoning part.
> - Using **standard zero-shot CoT** as the retrieval signal yields only modest gains and can even underperform at 6-shot, indicating that solution-oriented CoTs are not robust retrieval features and can be sensitive to noisy reasoning.
> - In contrast, both **retrieval-specific** prompts provide consistent improvements across all settings. This suggests that the structured, retrieval-specific CoT design is robust and still leads to higher-quality demonstrations overall. Furthermore, the math-specific retrieval prompt consistently outperforms the general one, indicating that tailoring the retrieval-specific CoT to the target domain can further enhance retrieval quality and downstream performance.
>
> Therefore, although the retrieval process is inherently noisy, our method consistently maintains strong performance, demonstrating robustness to retrieval imperfections and making it more applicable in realistic settings. We will incorporate these analyses into the revised paper.
>
> **You mention that a retrieval-specific CoT is produced for each test query, but it’s not fully clear whether this is generated by the same LMM used for final prediction, or a separate frozen module. Is it generated in a single pass? Is there any sampling or filtering involved? (Q1)**
>
> **A:** Thanks. We use the same LMM, such as Qwen2 VL 7B, for CoT generation, final prediction, with the retrieval-specific CoT being generated before retrieval. The CoT is generated in a **single pass** through the model, without any separate filtering or sampling procedure. During this process, we pass the test query (including the image and question) and retrieval-specific CoT prompt through the LMM, which outputs retrieval-specific CoT output.

---

> ### Author Response · Authors · 2025-11-24
> **Authors' Response (3/4)**
>
> **While the CoT-guided weighted retrieval (CWR) is well-motivated, the paper provides minimal insight into how the weighted combination of question-based and reasoning-based similarity actually impacts retrieval outcomes. The ablation on the weight parameter is helpful but not sufficient; qualitative examples or analysis of diversity versus redundancy among retrieved demonstrations would add depth. (W3)**
>
> **A:** Thanks. As discussed in the paper, the weighted combination of question-based and reasoning-based similarity is designed to balance the relevance of the test query and the reasoning cues. We further analyze this effect qualitatively in Appendix B.8, where we provide case studies under different values of the weight parameter.
>
> From these examples, we observe that increasing the weight on reasoning-based similarity tends to retrieve demonstrations with more similar solution strategies. In contrast, emphasizing question-based similarity yields examples that better match the query’s textual/visual surface form but may contain less informative or even irrelevant reasoning. For instance, in Figure 16, pure question-based retrieval returns demonstrations that do not involve the key triangle-sum criterion required by the test query and thus offer little help. These observations are consistent with our ablation on the weight parameter.
>
> **R4. On generalization beyond math and reproducibility (W5, W7, Q5)**
>
> **There is no evidence showing whether the model avoids overfitting or simply memorizes spurious patterns from the demonstrations. A comparison against simpler regularization strategies (e.g., dropout, prompt shuffling) could help clarify this. (W5)**
>
> **A:** Thanks for raising this concern. To examine whether our method overfits to specific demonstration patterns or prompt templates, we conducted a prompt-shuffling experiment on the MathVista 4-shot setting. Concretely, for each test instance we keep the retrieved demonstrations fixed but randomly permute their order in the context, and we repeat this procedure three times with different shuffles.
>
> **Table: Robustness to prompt shuffling on MathVista GPS (4-shot).**
>
> | Run  | Accuracy (%) |
> | ---- | ------------ |
> | 1    | 60.58        |
> | 2    | 61.06        |
> | 3    | 58.65        |
>
> The accuracy remains within a narrow range (58.65–61.06%) across runs, with fluctuations below 2.5 points. This stability suggests that the model is not simply memorizing a particular prompt pattern, but instead learns to robustly exploit the retrieved demonstrations.
>
> **Since the retrieval-specific CoT is explicitly math-oriented (extracting theorems, listing knowns, etc.), it is unclear whether the same method would work for general multimodal reasoning (e.g., VQA, diagram understanding). Would your method generalize to other multimodal reasoning domains like VQA, commonsense, or procedural understanding? What adaptations would be needed?  A small-scale test on a non-math domain would greatly strengthen the generalization claim. (W7, Q5)**
>
> **A:** Thanks. The core idea of CWR is to generate the deep reasoning behind test query to retrieve more relevant demonstrations, which is not limited to the mathematical domain. The math-oriented prompt shown in Figure 3 is one instantiation of this idea for mathematics task. We provide a more general, task-agnostic prompt in the appendix.
>
> To demonstrate the transferability of **CoT-MT³**, we further evaluate our method on GQA and M$^3$CoT using the general prompt, which focus on real-world VQA and multi-domain reasoning tasks, respectively. We randomly sample 200 examples from each dataset and report performance under different few-shot settings, as shown in the table below.
>
> **Table: Results on GQA and M$^3$CoT (200 randomly sampled examples)**
>
> | Method         | GQA (2-shot) | GQA (4-shot) | GQA (6-shot) | M$^3$CoT (2-shot) | M$^3$CoT (4-shot) | M$^3$CoT (6-shot) |
> | -------------- | ------------ | ------------ | ------------ | ----------------- | ----------------- | ----------------- |
> | **Zero-shot**  | 54.00        | 54.00        | 54.00        | 54.50             | 54.50             | 54.50             |
> | **QBICL**      | 59.00        | 55.50        | 54.00        | 56.50             | 56.00             | 57.00             |
> | **TTT-NN**     | 63.00        | 63.50        | 59.50        | 56.00             | 56.50             | 56.00             |
> | **TTT-ICL**    | 60.00        | 63.00        | 62.50        | 56.00             | 57.50             | 56.00             |
> | **CoT-MT$^3$** | **65.50**    | **64.50**    | **64.50**    | **57.50**         | **59.00**         | **59.50**         |
>
> The experimental results show that our method maintains strong performance on general benchmarks, demonstrating our method is simple yet generalizable. We will move these results into the main text and provide a more detailed analysis of the model’s performance on multimodal reasoning tasks.

---

> ### Author Response · Authors · 2025-11-24
> **Authors' Response (4/4)**
>
> **R5. On architectural details (W3, W6, W8, Q2, Q3, Q5, Q6)**
>
> **The retriever (Vista) is treated as a black box—its suitability for this specific task is not discussed. The similarity function is not formally defined—are image and text embeddings equally weighted, or are they learned separately? (W3, W6)**
>
> **For both question-based and reasoning-based similarity, what exactly is the feature encoder? Is it Vista, Qwen2-VL, or another model? Are the embeddings frozen or fine-tuned? Are text and image features handled jointly or separately? (Q2)**
>
> **You mention using MultiMath-300K for retrieval—do you use the entire corpus? How large is the retrieval index and how is it built?  What’s the average retrieval time and memory usage? (Q5)**
>
> **A:** Sorry for confusion. For both question-based and reasoning-based similarity, we use Vista as a frozen multimodal feature encoder. Vista takes the image and the associated text as joint input and produces a single multimodal embedding for each item, so image and text features are fused within Vista’s architecture. For question-based similarity, we jointly encode the test image and its textual question; for reasoning-based similarity, we encode the textual CoT cues produced for the test query.
>
> We build our retrieval index on top of these Vista embeddings using FAISS, and retrieve the top-$k$ demonstrations from this index. Concretely, we start from the full MultiMath-300K corpus and filter out items without the required bilingual statements, yielding 29.2K multimodal problems as our retrieval pool. FAISS-based retrieval adds only modest overhead: querying the index for top-$k$ neighbors takes on average 0.88 seconds per test instance on a single NVIDIA A800-80G GPU. We will report the index statistics and memory footprint in the paper.
>
> **Several important hyperparameters and architectural details are fixed without justification. For example, the LoRA configuration (rank 8, ) is not ablated. Why fix LoRA rank to 8 and $\alpha = 16$ ? It would be useful to know whether these settings are optimal. Did you experiment with other ranks or values of $\alpha$? Would full fine-tuning yield similar gains? An ablation here would help understand where the gains are coming from. (W6, Q3)**
>
> **A:** Thanks. The choice of LoRA rank 8 and $\alpha = 16$ is a commonly used and effective setting in LoRA-based fine-tuning, which balance performance and computational efficiency. We provide an ablation study of the Lora rank on the MathVista GPS subset (4-shot), as shown in following table:
>
> **Table: Ablation results of different Lora rank *r***
>
> | Method           | r = 8 | r = 16 | r = 32 |
> | ---------------- | ----- | ------ | ------ |
> | **MT$^3$(ours)** | 60.58 | 60.58  | 61.54  |
>
> Regarding full fine-tuning, our setting is per-sample test-time training, where updating all parameters would be unnecessarily heavy—incurring large computational and memory overhead given the very limited adaptation data. We will clarify this design choice in the paper.
>
> **The paper states that code will be released upon acceptance, but for a complex method combining retrieval, CoT generation, and fine-tuning, reproducibility is difficult without pseudo-code or algorithmic steps. A pseudocode version of MT$^3$ and CWR would really improve clarity and reproducibility. (W8, Q6)**
>
> **A:** Thank you once again for taking the time to review our paper! We will add a concise pseudocode (for CWR and MT$^3$)  in the Appendix. After the paper is accepted, we will release the code publicly to facilitate reproducibility. We also welcome questions and feedback from the community, and hope that our implementation can serve as a useful reference for future work!

---

> > ### Comment · Reviewer_CNNH · 2025-11-26
> >
> > Thank you for your rebuttal, indeed very interesting, but getting to the heart of the matter, I am confused about, the framework relies on the core assumption that "Retrieval-specific CoT can improve retrieval quality" (for example Table 4)
> >
> > I'd like to know if you have any way to prove that CoT to better retrieval to better adaptation is a causal chain, and not just that retrieval adds noise? This is because it directly impacts the contribution of the paper and is part of its integrity.
> >
> > BTW, have you conducted any causal intervention experiments?

---

> > > ### Author Response · Authors · 2025-11-28
> > > **Looking Forward to Your Reply**
> > >
> > > Thank you once again for taking the time to review our paper and for providing such insightful and constructive feedback.
> > >
> > > We have carefully considered each of your comments and have provided detailed responses. We sincerely hope that our efforts adequately address your concerns and contribute positively to your evaluation.
> > >
> > > As the author-reviewer discussion period concludes on Dec 03 (AoE), we would greatly appreciate any further feedback you may have. If you have any additional questions or require any clarifications, please do not hesitate to reach out to us.

---

> > ### Comment · Reviewer_CNNH · 2025-11-26
> >
> > For this part, My idea is more about "why Vista works", and your answer was mostly based on what Vista does. Regarding usability, your thoughts is very meaningful for this work.

---

> > > ### Author Response · Authors · 2025-11-27
> > > **Authors' Response to Reviewer CNNH**
> > >
> > > Thanks. We choose Vista because it provides a strong multimodal encoder that maps text and images into a shared representation space while supporting long textual inputs, which is crucial for our retrieval-specific CoT. In principle, we think CoT-MT$^3$ is compatible with any sufficiently strong multimodal encoder that can handle long text.

---

> ### Author Response · Authors · 2025-11-27
> **Authors' Response to Reviewer CNNH**
>
> Thank you once again for taking the time to review our work and for providing such insightful and constructive feedback, which is very helpful for improving our paper.
>
> In our framework, the retrieval-specific CoT is used only to compute retrieval scores and select few-shot demonstrations in CWR; it never participates in the final test-time adaptation. Consequently, its impact on downstream accuracy is entirely determines by the demonstration set it induces. In our setting, a retrieved example is considered “useful” if the model can make a correct prediction when using it as context.
>
> Therefore, to make the chain “CoT to better retrieval to better adaptation” more concrete, we evaluate retrieval in a pure ICL setting as follows:
>
> For each test instance, we retrieve the top-k examples from different retrieval strategies (Random, QB-Question based retrieval, CWR-CoT weighted retrieval) and use them as context for in-context learning. Since our retrieval corpus lacks relevance annotations, where standard IR metrics (e.g., recall@k) cannot be computed, the ICL accuracy serves as a practical proxy for retrieval usefulness. This setup acts as an intervention on the retrieval mechanism while keeping the adaptation procedure fixed, probing the “CoT to retrieval to adaptation” link. The results on MathVista GPS are:
>
> | #shots (k) | Zero-shot | Random + ICL | QB + ICL | CWR + ICL |
> | ---------- | --------- | ------------ | -------- | --------- |
> | 2-shot     | 46.15     | 42.31        | 48.56    | **49.52** |
> | 4-shot     | 46.15     | 40.87        | 46.63    | **48.56** |
> | 6-shot     | 46.15     | 39.42        | 45.67    | **51.92** |
>
> From this table, we have the following observations:
>
> 1. When randomly injecting demonstrations, performance degrades as $k$ increases, whereas both QB and CWR avoid this degradation. This shows that **retrieval quality directly affects adaptation**, i.e., informative demonstrations are crucial for reasoning, while random ones act as harmful noise.
> 2. Across all $k \in {2,4,6}$, CWR consistently outperforms QB, with the largest gain at 6-shot. This indicates that **CoT-guided retrieval systematically provides more useful demonstrations than query-only retrieval, resulting in better adaptation**, and that this advantage becomes even more pronounced as more demonstrations are used. In other words,  **retrieval-specific CoT indeed leads to better retrieval behavior (and then better reasoning performance) across different $k$**.
>
> We will integrate this analysis into the revised version to more clearly support the intended chain “CoT to better retrieval to better adaptation”. If you still have any further questions, please don't hesitate to discuss with us. Thank you very much!

---

### Official Review · Reviewer_88T1 · 2025-10-25

**Soundness:** 2
**Presentation:** 3
**Contribution:** 2
**Rating:** 4
**Confidence:** 3

**Summary:**

This paper proposed a retrieval-augmented framework to improve few-shot multimodal reasoning at test time. It has two pieces: (1) CoT-Guided Weighted Retrieval (CWR), which first prompts the model to produce a retrieval-specific CoT and then retrieves demonstrations using a weighted combination of query similarity and reasoning-similarity; (2) Meta Test-Time Training (MT3), which fine-tunes the model on multiple few-shot variants (varying context sizes and combinations) constructed from the retrieved demos to improve robustness.

**Strengths:**

* CoT-Guided Retrieval. The paper proposed a novel CoT-guided retrieval approach that enhances relevance compared to standard query-document matching.
* Retrieval-Specific CoT Prompt. The authors designed a retrieval-specific CoT prompt to extract key reasoning information, addressing the limitation of conventional CoT prompts that only focus on the question itself.
* Thorough ablation studies. The paper conducts thorough ablation studies to disentangle the contributions of different components, including the retrieval weight w and context size k. Results show that both CWR and MT³ bring significant gains, the optimal performance achieved at a retrieval weight of $w = 0.7$, a practical recommendation of setting $k = \lfloor m/2 \rfloor$.

**Weaknesses:**

* Limited Evaluation Breadth. While the paper provides extensive experiments on multi-modal math reasoning tasks, it does not evaluate the generalizability of CoT-MT³ to other multi-modal tasks. Whether CoT-MT³ can transfer effectively to broader multi-modal settings remains an open question.
* Gains Are Uneven and Sometimes Marginal. As shown in Table 1, CoT-MT³ does not consistently outperform all baselines. For instance, it is not the best-performing method under the TD (2-shot) setting. Moreover, the improvements are relatively small in several configurations, such as TD (6-shot), TL (4-shot), VI (4-shot), and the overall 2-shot and 4-shot results.
* Lack of Statistical Significance. In cases where the performance differences are minor (as in Table 1), the paper should report statistical significance to confirm that the observed gains are attributable to CoT-MT³ rather than random variation.
* Compute Cost and Practicality of Test-Time Fine-Tuning. The method involves fine-tuning per test query with multiple prompt variants, which may limit its applicability in real-time or interactive scenarios. The paper lacks a discussion on latency and computational cost, which are essential for assessing the practicality of the approach.

**Questions:**

1.	What is the average MT³ time consumption per query?
2.	Are the choices of $w = 0.7$ and $k = \lfloor m/2 \rfloor$ universally optimal across different datasets?
3.	Are the observed gains statistically significant when the improvements are marginal?
4.	Can CoT-MT³ be applied to other types of multi-modal tasks beyond science/math domains?

---

> ### Author Response · Authors · 2025-11-24
> **Authors' Response (1/2)**
>
> Thanks for your insightful comments and suggestions.
>
> **W1. [While the paper provides extensive experiments on multi-modal math reasoning tasks, it does not evaluate the generalizability of CoT-MT³ to other multi-modal tasks. Whether CoT-MT³ can transfer effectively to broader multi-modal settings remains an open question.]**
>
> **A:** Thanks. The complex reasoning in the mathematical domain is the most challenging in multimodal reasoning. As the first work on TTT for multimodal reasoning, we thus focus on this challenging problem, but our proposed method is not limited to the mathematical domain.
>
> To demonstrate the transferability of CoT-MT³, we evaluated its performance on GQA and M$^3$CoT, which target general visual question answering (VQA) and multi-domain complex reasoning, respectively. We randomly sampled 200 examples and report the results across different few-shot settings, as shown in the table below:
>
> **Table: Results on GQA and M$^3$CoT (200 randomly sampled examples)**
>
> | Method         | GQA (2-shot) | GQA (4-shot) | GQA (6-shot) | M$^3$CoT (2-shot) | M$^3$CoT (4-shot) | M$^3$CoT (6-shot) |
> | -------------- | ------------ | ------------ | ------------ | ----------------- | ----------------- | ----------------- |
> | **Zero-shot**  | 54.00        | 54.00        | 54.00        | 54.50             | 54.50             | 54.50             |
> | **QBICL**      | 59.00        | 55.50        | 54.00        | 56.50             | 56.00             | 57.00             |
> | **TTT-NN**     | 63.00        | 63.50        | 59.50        | 56.00             | 56.50             | 56.00             |
> | **TTT-ICL**    | 60.00        | 63.00        | 62.50        | 56.00             | 57.50             | 56.00             |
> | **CoT-MT$^3$** | **65.50**    | **64.50**    | **64.50**    | **57.50**         | **59.00**         | **59.50**         |
>
> The experimental results show that our method maintains strong performance on general benchmarks, demonstrating our method is simple yet effective and generalizable. We will move these results into the main text and provide a more detailed analysis of the model’s performance on multimodal reasoning tasks.
>
> **W2. [As shown in Table 1, CoT-MT³ does not consistently outperform all baselines. For instance, it is not the best-performing method under the TD (2-shot) setting. Moreover, the improvements are relatively small in several configurations, such as TD (6-shot), TL (4-shot), VI (4-shot), and the overall 2-shot and 4-shot results.]**
>
> **A:** Thanks for your observation. CoT-MT$^3$ does not achieve the best result in every single setting. However, it remains competitive with, or stronger than, the strongest baseline in almost all settings. Even in the configurations where the absolute gains appear small (e.g., TD 6-shot, TL 4-shot, VI 4-shot), CoT-MT³ remains slightly better than the strongest baseline, which is non-trivial given the difficulty of these multimodal reasoning tasks.
>
> As shown in Tables 1, 2, and 3, when averaged over all shot settings, CoT-MT$^3$ improves over the strongest baseline by **4.82** points on MathVerse, **5.13** points on MathVista-GPS, and **2.86** points on We-Math. When averaged over all datasets, CoT-MT$^3$ further improves over the strongest baseline by **2.80** points in the 2-shot setting, **4.64** points in the 4-shot setting, and **2.69** points in the 6-shot setting, indicating that the method is generally effective rather than tuned to a specific dataset or configuration. Moreover, compared to the zero-shot baseline, CoT-MT$^3$ achieves at least a **3-point** improvement in **18 out of 21** few-shot settings. Given the strength of the existing baselines and the difficulty of these benchmarks, such consistent gains are meaningful. We will clarify this emphasis on robustness across benchmarks and settings in the revised version.
>
> **W3. [In cases where the performance differences are minor (as in Table 1), the paper should report statistical significance to confirm that the observed gains are attributable to CoT-MT³ rather than random variation.]**
>
> **A:** Thank you for the suggestion. We have conducted an additional significance analysis on the MathVista (GPS) benchmark. For each method and each few-shot setting (2-/4-/6-shot), we run three independent trials and record the number of correctly answered questions. We then compute the mean ± standard deviation and perform a two-sided *t*-test between CoT-MT$^3$ and the strongest baseline in each setting.
>
> **Table: Statistical comparison on MathVista (GPS) over 3 runs.**
>
> | Method     | 2-shot           | 4-shot           | 6-shot           |
> | ---------- | ---------------- | ---------------- | ---------------- |
> | QB-ICL     | 51.60 ± 1.21     | 47.12 ± 0.48     | 49.84 ± 1.94     |
> | TTT-ICL    | 51.12 ± 1.11     | 55.29 ± 1.27     | 53.69 ± 1.00     |
> | CoT-MT$^3$ | **56.73 ± 0.48** | **60.42 ± 1.21** | **59.62 ± 1.44** |

---

> ### Author Response · Authors · 2025-11-24
> **Authors' Response (2/2)**
>
> Across the 2-, 4-, and 6-shot settings, CoT-MT$^3$ consistently achieves higher mean accuracy than other methods. Based on the three paired runs in each setting, a paired *t*-test shows that all improvements are statistically significant (*p* < 0.05). We will incorporate these statistical results into the revised version of the paper and continue to extend this analysis to additional settings to further strengthen the empirical rigor.
>
> **W4. [The method involves fine-tuning per test query with multiple prompt variants, which may limit its applicability in real-time or interactive scenarios. The paper lacks a discussion on latency and computational cost, which are essential for assessing the practicality of the approach.]**
>
> **A:** Thanks for pointing out the importance of latency and computational cost. We report a detailed latency analysis of total latency and computational cost in Appendix B.5, and we will move a summary of these results into the main paper in the revision version. Concretely, Table 4 compares average accuracy and GPU time per test sample for different TTT methods. The table 4 is as follows:
>
> **Table: Training overhead of different TTT methods per query**
>
> | GPU time (mins)  | 2-shot | 4-shot | 6-shot |
> | ---------------- | ------ | ------ | ------ |
> | TTT-NN           | 0.1115 | 0.1258 | 0.1867 |
> | TTT-ICL          | 0.1044 | 0.1313 | 0.1892 |
> | **MT$^3$(ours)** | 0.1168 | 0.1539 | 0.1913 |
>
> CoT-MT$^3$ introduces only a very small additional training overhead compared to existing TTT methods, while yielding clearly higher accuracy. Additionally, Figure 10 plots the accuracy–latency trade-off for all methods. CoT-MT$^3$ lies on the best part of this curve, continuing to benefit from additional compute rather than saturating early. Given that our focus lies on **accuracy-critical** scenarios, especially for complex reasoning, we view a modest increase in per-query compute is acceptable in exchange for significantly improved multimodal reasoning. We will move these analyses to the main text.
>
> **Q1. [What is the average MT³ time consumption per query?]**
>
> **A:** Thanks. We have explained it in our response to Weakness 4.
>
> **Q2. [Are the choices of $w=0.7$  and $k=m/2$ universally optimal across different datasets?]**
>
> **A:** Thanks for pointing this out. Our choices of $w = 0.7$ and $k = m/2$ are empirical and intended as robust default settings. In practice, the optimal values may vary across different queries. For example, when the test query contains rich, detailed descriptions, a larger $w$ may be beneficial; conversely, when the question is vague, a smaller $w$ can help the model rely more on the reasoning-based similarity from CoT.
>
> In our experiments, we found that these default values strike a good balance between stability and performance. We provide additional ablation studies on Mathverse TD as follows:
>
> **Table: Ablation results for different k values for 4-shot setting**
>
> | Dataset         | k=0            | k=1   | k=2       | k=3       |
> | --------------- | -------------- | ----- | --------- | --------- |
> | MathVerse (TD)  | 38.96          | 39.59 | **40.36** | 38.70     |
> | MathVista (GPS) | 54.81          | 55.29 | **60.58** | 58.17     |
>
> **Table: Ablation results for different w values for 4-shot setting**
>
> | Dataset         | w=0.1          | w=0.3 | w=0.5 | w=0.7     | w=0.9     | w=1   |
> | --------------- | -------------- | ----- | ----- | --------- | --------- | ----- |
> | MathVerse (TD)  | 36.92          | 37.44 | 38.83 | **40.36** | 39.47     | 36.68 |
> | MathVista (GPS) | 49.52          | 52.40 | 53.85 | **60.58** | 55.78     | 52.88 |
>
> As shown in our ablation studies, the chosen settings yield competitive results **without per-dataset fine-tuning**. While we do not claim these choices are optimal for all scenarios, they serve as strong and practical defaults. We will incorporate these results in paper.
>
> **Q3. [Are the observed gains statistically significant when the improvements are marginal?]**
>
> **A:** Thanks. We have explained it in our response to Weakness 3.
>
>
> **Q4. [Can CoT-MT³ be applied to other types of multi-modal tasks beyond science/math domains?]**
>
> **A:** Thanks. We have explained it in our response to Weakness 1.

---

> ### Author Response · Authors · 2025-11-27
> **Looking Forward to Your Reply**
>
> Thank you once again for taking the time to review our paper and for providing such insightful and constructive feedback.
>
> We have carefully considered each of your comments and have provided detailed responses. We sincerely hope that our efforts adequately address your concerns and contribute positively to your evaluation.
>
> As the author-reviewer discussion period concludes on Dec 03 (AoE), we would greatly appreciate any further feedback you may have. If you have any additional questions or require further clarifications, please do not hesitate to discuss with us.

---

> > ### Comment · Reviewer_88T1 · 2025-11-28
> >
> > Thank the authors for providing evidence to show the generalizability and latency. I would like to slightly increase my score to 6. But I cannot operate it on the system. I hope the AC can notice that.

---

> > > ### Author Response · Authors · 2025-11-28
> > > **Authors' Response to Reviewer 88T1**
> > >
> > > We sincerely appreciate you raising your score and are truly grateful for your constructive feedback, as well as the time you've dedicated to improving the quality of our work. Please don't hesitate to let us know if there's anything further to discuss. Thank you very much again!

---

### Official Review · Reviewer_m2RX · 2025-10-28

**Soundness:** 3
**Presentation:** 3
**Contribution:** 2
**Rating:** 4
**Confidence:** 4

**Summary:**

The paper proposes CoT-MT3, a framework for few-shot multimodal reasoning that combines CoT-guided weighted retrieval (CWR) and meta test-time training (MT3). CWR enhances retrieval relevance by generating a structured, retrieval-specific chain-of-thought to capture key problem information and reasoning semantics. MT3 improves robustness by meta-training the model on diverse few-shot contexts of varying sizes during test time. The method demonstrates consistent gains across multiple multimodal mathematical reasoning benchmarks and exhibits robustness to context length variations. The experimental design is thorough and the ablation studies are informative.

**Strengths:**

1. The paper has good experimental design, technical implementation, and result analysis. It conducts evaluations on three mathematical benchmarks, covering different modality dependence levels and diagnostic metrics.
2. The writting in this paper is clear, which makes the method easy to understand and reproduce.

**Weaknesses:**

1. It lacks direct evaluation of retrieval quality. The paper only reports downstream reasoning accuracy but never directly measures whether the CoT-Guided Weighted Retrieval (CWR) actually retrieves more "reasoning-aligned" examples.
2. It over-relies on math-specific structures, which limits its generalization claims. The retrieval-specific Chain-of-Thought (CoT) prompt (Figure 3) is explicitly designed for mathematics. Therefore, the paper’s claim of advancing "multimodal reasoning" is at risk of overgeneralization.
3. The core idea of "retrieval-specific CoT", i.e., prompting the model to first generate a preliminary solution plan (identifying key infomation, theorems, and reasoning steps) and then using this plan to retrieve demonstrations with similar reasoning paths, has been explored in existing ICL literature[1]. Therefore, this strategy indeed  lacks significant novelty. Furthermore, the paper fails to provide an ablation study on the four distinct components of its retrieval-specific prompt (Figure 3), making it unclear which of these components are critical for the retrieval improvement and whether the complex prompt design is fully justified.

[1] In-context learning with iterative demonstration selection.

**Questions:**

1. The paper uses a fixed weight w = 0.7 across all benchmarks and shot settings (Section 4.1). Is this value robust across different domains and model scales?
2. The retrieval-specific Chain-of-Thought (CoT) prompt (Figure 3) is explicitly designed for mathematical reasoning ("Identify relevant mathematical theorems"). Is this method still effective in non-mathematical domains?
3. The paper reports downstream reasoning accuracy but does not directly evaluate retrieval relevance. What proportion of the top-m retrieved examples are reasoning-aligned (use the same theorems or strategies)?

---

> ### Author Response · Authors · 2025-11-24
> **Authors' Response (1/3)**
>
> **W1. [It lacks direct evaluation of retrieval quality. The paper only reports downstream reasoning accuracy but never directly measures whether the CoT-Guided Weighted Retrieval (CWR) actually retrieves more "reasoning-aligned" examples.]**
>
> **A:** Thanks for your insightful review. Our retrieval corpus and test set come from existing benchmarks and training data, which don't provide any relevance annotations linking retrieved examples to specific test queries. Therefore, traditional IR metrics such as precision/recall@k cannot be computed in our setting. In our task, an example is considered “useful” if the model can make correct predictions when using it as context. For each test instance, we retrieve the top-k examples using either standard query-based retrieval (QB) or CWR and use them as context for in-context learning. The result on MathVista GPS is as follows:
>
> **Table: The analysis of retrieval quality**
>
> | #shots (k) | QB    | CWR       |
> | ---------- | ----- | --------- |
> | 2-shot     | 48.56 | **49.52** |
> | 4-shot     | 46.63 | **48.56** |
> | 6-shot     | 45.67 | **51.92** |
>
> As shown in table, CWR consistently outperforms standard query-based retrieval, with the largest gain observed at 6-shot. This pattern indicates that CWR consistently retrieves examples that are more useful for reasoning, and thus more likely to be “reasoning-aligned” in practice.
>
> We also provide visualizations of the retrieved demonstrations in Appendix B.8 to give a clearer picture of how CWR affects the actual retrieval behavior. We observe that incorporating reasoning-based similarity tends to retrieve demonstrations whose solution strategies more closely match the ground-truth CoT. We will integrate this analysis into the revised paper.
>
> **W2. [It over-relies on math-specific structures, which limits its generalization claims. The retrieval-specific Chain-of-Thought (CoT) prompt (Figure 3) is explicitly designed for mathematics. Therefore, the paper’s claim of advancing "multimodal reasoning" is at risk of overgeneralization.]**
>
> **A:** Thanks. The core idea of CWR is to generate the deep reasoning behind test query to retrieve more relevant demonstrations, which is not limited to the mathematical domain. The math-oriented prompt shown in Figure 3 is one instantiation of this idea for mathematics task. In practice, CWR can be flexibly adapted to produce customized reasoning cues for different task types. We provide a more general, task-agnostic prompt in the Appendix.
>
> We further evaluate our method on GQA and M$^3$CoT, which focus on real-world VQA and multi-domain reasoning tasks, respectively. For these general benchmarks, we use the general retrieval-specific CoT prompt, randomly sample 200 examples from each benchmark and report performance under different few-shot settings, as shown in the table below.
>
> **Table: Results on GQA and M$^3$CoT (200 randomly sampled examples)**
>
> | Method         | GQA (2-shot) | GQA (4-shot) | GQA (6-shot) | M$^3$CoT (2-shot) | M$^3$CoT (4-shot) | M$^3$CoT (6-shot) |
> | -------------- | ------------ | ------------ | ------------ | ----------------- | ----------------- | ----------------- |
> | **Zero-shot**  | 54.00        | 54.00        | 54.00        | 54.50             | 54.50             | 54.50             |
> | **QBICL**      | 59.00        | 55.50        | 54.00        | 56.50             | 56.00             | 57.00             |
> | **TTT-NN**     | 63.00        | 63.50        | 59.50        | 56.00             | 56.50             | 56.00             |
> | **TTT-ICL**    | 60.00        | 63.00        | 62.50        | 56.00             | 57.50             | 56.00             |
> | **CoT-MT$^3$** | **65.50**    | **64.50**    | **64.50**    | **57.50**         | **59.00**         | **59.50**         |
>
> The experimental results show that our method maintains strong performance on general benchmarks, demonstrating our method is simple yet effective and generalizable. We will move these results into the main text and provide a more detailed analysis of the model’s performance on general reasoning tasks.

---

> ### Author Response · Authors · 2025-11-24
> **Authors' Response (2/3)**
>
> **W3. [The core idea of "retrieval-specific CoT", i.e., prompting the model to first generate a preliminary solution plan (identifying key infomation, theorems, and reasoning steps) and then using this plan to retrieve demonstrations with similar reasoning paths, has been explored in existing ICL literature[1]. Therefore, this strategy indeed lacks significant novelty. Furthermore, the paper fails to provide an ablation study on the four distinct components of its retrieval-specific prompt (Figure 3), making it unclear which of these components are critical for the retrieval improvement and whether the complex prompt design is fully justified.]**
>
> **A:** We thank the reviewer for the insightful comments. Our explnation is two fold:
>
> (1) Using intermediate reasoning as retrieval signal is a natural solution in ICL. IDS [1] leverages few-shot CoT outputs to iteratively retrieve new demonstrations and update the in-context reasoning outputs. However, IDS never generate a preliminary solution plan but using standard zero-shot/few-shot CoT to retrieve and reasoning, which leads to unstable retrieval signals.
>
> In contrast, our retrieval-specific CoT is explicitly designed for retrieval rather than answering. The prompt guides the model to decompose the solution into structured sub-tasks (e.g., key information, relevant knowledge), which is more helpful for retrieval. This retrieval-specific, structured CoT design, which integrated into our CWR / CoT-MT$^3$ framework and validated on both math and multimodal benchmarks, isn't explored in IDS. We will clarify this distinction in the revised version.
>
> (2) On the ablation study of the retrieval-specific CoT prompt.
>
> For a deeper analysis of the effectiveness of prompt design, we conducted an ablation study on different CoT prompt types on the MathVista GPS. Concretely, we compare:
>
> (1) the original zero-shot prompt (as shown in Figure 9 of the paper), (2) the general Retrieval-Specific CoT prompt (as shown in the Appendix B.4), and (3) the Retrieval-Specific CoT prompt tailored for mathematics (as shown in Figure 3 of the paper). The results are as follows:
>
> **Table: the ablation study of different CoT prompt of CoT-MT$^3$ on MathVista (GPS)**
>
> | CoT Prompt Type                  | 2-shot    | 4-shot    | 6-shot    |
> | -------------------------------- | --------- | --------- | --------- |
> | W/o CoT (Query-Based Retrieval)  | 50.96     | 55.77     | 56.73     |
> | Zero-shot CoT                    | 55.77     | 56.73     | 56.25     |
> | Retrieval-Specific CoT (General) | **57.21** | 59.13     | 57.21     |
> | Retrieval-Specific CoT (Math)    | **57.21** | **60.58** | **58.65** |
>
> We observe three consistent trends:
>
> - **CoT vs. no CoT.** Introducing any CoT already yields a large improvement over “W/o CoT (Query-Based Retrieval)”, showing that incorporating explicit reasoning is important for effective retrieval.
> - **Retrieval-specific vs. generic CoT.** Using standard zero-shot CoT as the retrieval signal yields only modest gains and can even underperform at 6-shot, indicating that solution-oriented CoTs are not robust retrieval features and can be sensitive to noisy reasoning. However, the general retrieval-specific CoT significantly outperforms the standard zero-shot CoT (e.g., 4-shot: 56.73 → 59.62). This demonstrates that a CoT explictly designed for retrieval provides additional benefits beyond a generic solution-oriented CoT.
>
> - **Task-tailored retrieval-specific CoT design.** The math retrieval-specific CoT further improves over the general one and achieves the best performance, suggesting that retrieval-specific CoT design can be task-tailored which leads to consistently better retrieval and reasoning performance.
>
> These results support that our retrieval-specific CoT design is both effective and justified: more structured and task-tailored CoT prompts lead to consistently better retrieval and reasoning performance. We will incorporate this analysis into our revision version.
>
> [1] In-Context Learning with Iterative Demonstration Selection.

---

> ### Author Response · Authors · 2025-11-24
> **Authors' Response (3/3)**
>
> **Q1. [The paper uses a fixed weight w = 0.7 across all benchmarks and shot settings (Section 4.1). Is this value robust across different domains and model scales?]**
>
> **A:** Thanks. It is generally unrealistic to find a universal optimal hyperparameter across all possible domains and model scales.  The weight $w$ is introduced to balance the contribution of the test query and the generated CoT cues in retrieval. We find that $w = 0.7$ works as a robust and simple setting across complex reasoning benchmarks.
>
> However, this balance could be further refined across different domains. For example, in some special tasks such as chart understanding, the question itself is already very informative and a larger $w$ would be more appropriate, in such cases, $w$ should be increased. In other cases, the test query may induce noisy retrieval, whereas the model-generated CoT cues offer a cleaner reformulation; and a smaller $w$ would be more appropriate.
>
> **Q2. [The retrieval-specific Chain-of-Thought (CoT) prompt (Figure 3) is explicitly designed for mathematical reasoning ("Identify relevant mathematical theorems"). Is this method still effective in non-mathematical domains?]**
>
> **A:** Thanks. We have explained it in our response to Weakness 2.
>
> **Q3. [The paper reports downstream reasoning accuracy but does not directly evaluate retrieval relevance. What proportion of the top-m retrieved examples are reasoning-aligned (use the same theorems or strategies)?]**
>
> **A:** Thanks. We have explained it in our response to Weakness 1.

---

> > ### Comment · Reviewer_m2RX · 2025-11-26
> >
> > Thanks for the rebuttal. The authors have addressed all my concerns. Therefore, I will increase the score accordingly.

---

> > > ### Author Response · Authors · 2025-11-27
> > > **Authors' Response to Reviewer m2RX**
> > >
> > > Thank you for your kind response and for considering adjusting the score. We sincerely appreciate your insightful review and the time you've dedicated to improving the quality of our work. Please don't hesitate to let us know if there's anything further to discuss.

---

### Official Review · Reviewer_wwJS · 2025-10-31

**Soundness:** 2
**Presentation:** 4
**Contribution:** 2
**Rating:** 4
**Confidence:** 4

**Summary:**

This work proposes CoT-MT^3, an approach which targets to improve LLMs' multimodal reasoning capabilities in retrieval augmented few-shot learning settings. The proposed methodology combines CoT and test-time scaling. First, the proposed methodology implements a CoT-guided strategy to retrieve relevant examples instead of retrieving them conditioned directly on question query only. The second stage, meta test-time scaling, finetunes the large multimodal model (LLM) on a small set of training data which consists of retrieved few-shot examples with different number of demonstrations. The proposed approach is tested on 3 distinct mathematical benchmarks where CoT-MT^3 significantly outperforms previous approaches. The ablation studies cover investigating the contribution of different components, and some hyperparameters.

**Strengths:**

- Simple yet effective approach. Implementing CoT-MT^3 should be appropriately easy for an average researcher. However, it significantly outperforms previous methodologies.
- The paper will focuses on a currently active and impactful direction.
- Good presentation. The paper is well written, easy to understand, and it contains good self-explanatory figures.

**Weaknesses:**

- The evaluations only cover multimodal mathematical reasoning benchmarks. Nonetheless, the title and abstract are misleading in this sense. There are experiments on GQA dataset, yet represented in the appendix. It'd be good to push this part to the main text, even include more reasoning tasks with multimodal signals. Another option is to extend limitations section.
- The experiments only cover one vision-language model with 7B parameters (`Qwen2-VL-7B`). It would be interesting to see the significance of model scale (both larger and smaller) and different families (e.g., `Qwen2.5-VL`, `Idefics3`)
- The ablation studies lack an important study, where one could combine question-based querying (QB) with the proposed meta test-time training approach. Fig. 4 illustrates that that MT^3's contribution is much more larger than CoT-based retrieval.
- There are no ablation studies examining the chain-of-thought prompt.

**Questions:**

- Do you think that the approach could benefit from a per-sample adaptive `w` argument, where `w` takes a different value depending on the question?
- Could you also please share the exact prompt used for GQA evalautions? I think it should be different than the prompt used for other benchmarks, as the prompt Fig. 3 starts with *You are a mathematics expert. I will now provide you with a multimodal math problem.*.

---

> ### Author Response · Authors · 2025-11-24
> **Authors' Response (1/3)**
>
> Thank you for your insightful comments and questions.
>
> **W1. [The evaluations only cover multimodal mathematical reasoning benchmarks. Nonetheless, the title and abstract are misleading in this sense. There are experiments on GQA dataset, yet represented in the appendix. It'd be good to push this part to the main text, even include more reasoning tasks with multimodal signals. Another option is to extend limitations section.]**
>
> **A:** Thank you for pointing this out. The complex reasoning in the mathematical domain is the most challenging in multimodal reasoning. As the first work on TTT for multimodal reasoning, we thus focus on this challenging problem, but our proposed method is not limited to the mathematical domain.
>
> Besides GQA, we also evaluate on the M$^3$CoT [1], which focus on multi-domain complex reasoning. Similarly to GQA, we randomly sample 200 examples and report the results under different few-shot settings, as shown in the table below.
>
> **Table: Results on GQA and M$^3$CoT (200 randomly sampled examples)**
>
> | Method         | GQA (2-shot) | GQA (4-shot) | GQA (6-shot) | M$^3$CoT (2-shot) | M$^3$CoT (4-shot) | M$^3$CoT (6-shot) |
> | -------------- | ------------ | ------------ | ------------ | ----------------- | ----------------- | ----------------- |
> | **Zero-shot**  | 54.00        | 54.00        | 54.00        | 54.50             | 54.50             | 54.50             |
> | **QBICL**      | 59.00        | 55.50        | 54.00        | 56.50             | 56.00             | 57.00             |
> | **TTT-NN**     | 63.00        | 63.50        | 59.50        | 56.00             | 56.50             | 56.00             |
> | **TTT-ICL**    | 60.00        | 63.00        | 62.50        | 56.00             | 57.50             | 56.00             |
> | **CoT-MT$^3$** | **65.50**    | **64.50**    | **64.50**    | **57.50**         | **59.50**         | **59.50**         |
>
>
> The experimental results show that our method still maintains strong performance on general benchmarks, demonstrating our method is simple yet generalizable. We will move these results into the main text and provide a more detailed analysis of the model’s performance on multimodal reasoning tasks. And we will extend the limitations section to more clearly explain the scope of our current evaluation and the generality of the method.
>
> [1] M$^3$CoT: A Novel Benchmark for Multi-Domain Multi-step Multi-modal Chain-of-Thought. ACL 2024.
>
> **W2. [The experiments only cover one vision-language model with 7B parameters (Qwen2-VL-7B). It would be interesting to see the significance of model scale (both larger and smaller) and different families (e.g., Qwen2.5-VL, Idefics3)]**
>
> **A:** Thanks. To evaluate the impact of model scale and backbone family, we additionally conduct experiments with a smaller model, **Qwen2-VL-2B**, and a larger model from a different family, **Pixtral-12B**, on the MathVista (GPS) subset under 2- and 4-shot settings.
> We provided a more detailed explanation in the Appendix B.5.1. The results are shown below:
>
> **Table: Accuracy (%) of different backbone–method combinations on MathVista (GPS)**
>
> | Shots  | Backbone    | Zero-shot | QBICL | TTT-NN | TTT-ICL | CoT-MT$^3$ |
> | ------ | ----------- | :-------: | :---: | :----: | :-----: | :--------: |
> | 2-shot | Qwen2-VL-2B |   37.98   | 39.90 | 33.65  |  40.87  | **44.23**  |
> |        | Pixtral-12B |   39.90   | 48.56 | 44.71  |  51.92  | **52.40**  |
> | 4-shot | Qwen2-VL-2B |   37.98   | 40.87 | 40.38  |  40.87  | **42.79**  |
> |        | Pixtral-12B |   39.90   | 51.44 | 49.04  |  48.56  | **52.88**  |
>
> The experimental results show that our method maintains strong performance across different backbone models. This aligns with our results on Qwen2-VL-7B and suggests that our method is model-agnostic and works well for LMMs of varying sizes and families.

---

> > ### Comment · Reviewer_wwJS · 2025-11-26
> >
> > I thank the authors for their response. Would it be possible to repeat some of these experiments with instruction-tuned versions of these models also as well, to see whether how much instruction-tuned and base pretrained models benefit from the proposed approach?
> >
> > For the ablations on the prompt: I was asking more about the components in the prompt as it has 4 numbered bullets. For instance, one could remove _Identify the key information and unknowns:_ part and then observe how results vary in that case.

---

> > > ### Author Response · Authors · 2025-11-28
> > > **Authors' Response to Reviewer wwJS**
> > >
> > > Thank you once again for taking the time to review our work and for providing such insightful and constructive feedback, which is very helpful for improving our paper.
> > >
> > > **Q1 [Would it be possible to repeat some of these experiments with instruction-tuned versions of these models also as well, to see whether how much instruction-tuned and base pretrained models benefit from the proposed approach?]**
> > >
> > > **A:**   Thank you for the question. In this study, we use instruction-tuned versions of the models (e.g., Qwen2-VL-7B-Instruct) rather than the base pretrained models. Sorry for the misunderstanding and we will clarify this in the revision.
> > >
> > > In practice, most recent open-source large multimodal models (LMMs) are released only in their instruction-tuned versions, with no publicly available base pretrained checkpoints. Regardless of this, in order to evaluate the performance of our approach on base pretrained models, we conduct experiments with Qwen2-VL-7B. Due to its limited instruction-following ability, even with the initial zero-shot CoT prompt, the model performs poorly on complex reasoning tasks and frequently produces repetitive outputs. We thus focus on the evaluation with instruction-tuned LMMs. However, we believe our approach is also applicable to base pretrained models with stronger zero-shot abilities.
> > >
> > > **Q2 [For the ablations on the prompt: I was asking more about the components in the prompt as it has 4 numbered bullets. For instance, one could remove *Identify the key information and unknowns:* part and then observe how results vary in that case.]**
> > >
> > > **A:** Thanks for your clarification. We conduct an ablation study on the retrieval-specific CoT prompt by removing one subtask at a time. The results on MathVista GPS (4-shot) are presented below:
> > >
> > > **Table: Ablation of Retrieval-specific CoT prompt**
> > >
> > > | **Variant (Removed Component)**                              | **Accuracy (%)** |
> > > | ------------------------------------------------------------ | ---------------- |
> > > | **Complete Prompt**                                          | **60.58**        |
> > > | W/o Subtask 1: “Understand problem and list given information” | 54.81 (-5.77)    |
> > > | W/o Subtask 2: “Identify key information and unknowns”       | 57.69 (-2.89)    |
> > > | W/o Subtask 3: “Identify relevant mathematical theorems”     | 58.17 (-2.41)    |
> > > | W/o Subtask4: “Step‑by‑step reasoning based on understanding” | 57.12 (-3.46)    |
> > >
> > > The experimental results show that removing any subtask of retrieval-specific CoT consistently degrades the performance. In particular, we observe that:
> > >
> > > 1. Removing subtask 1 (**problem understanding**) leads to a significant drop (5.77%) in final performance. This subtask explicitly guides the model to identify and list the key information given in the problem, thereby **producing the core understanding of the query and serving as the foundation for all subsequent subtasks**. This supports our claim that retrieval-specific CoT should be grounded in an explicit deep understanding of the test query in order to improve retrieval quality and subsequent reasoning performance.
> > > 2. By checking the CoT outputs generated in the ablation settings, we observe that the model sometimes still produces content corresponding to the removed subtasks (e.g., it may output step-by-step reasoning or a final answer even after we remove subtask 4).  Nevertheless, the overall performance consistently degrades once these subtasks are removed from the prompt. This demonstrates that **an explicitly structured, retrieval-specific CoT design** is necessary for achieving better model performance.
> > >
> > > We will incorporate these results into the revision and clarify the model version we use. If you still have any questions, please don't hesitate to further discuss with us. Thank you very much again!

---

> ### Author Response · Authors · 2025-11-24
> **Authors' Response (2/3)**
>
> **W3. [The ablation studies lack an important study, where one could combine question-based querying (QB) with the proposed meta test-time training approach. Fig. 4 illustrates that that MT^3's contribution is much more larger than CoT-based retrieval.]**
>
> **A:** Thanks. We conduct the additional ablation study that combines question-based retrieval (QB) and our proposed MT$^3$. The complete ablation results are shown as below:
>
> **Table: Ablation of different components of CoT-MT$^3$ on MathVista (GPS)**
>
> | Method           | Accuracy(%) |
> | ---------------- | ----------- |
> | Zero-shot        | 46.15       |
> | QB + ICL         | 46.53       |
> | CWR + ICL        | 48.56       |
> | CWR + TTT-NN     | 52.40       |
> | CWR + TTT-ICL    | 51.44       |
> | *QB + MT$^3$*    | 55.77       |
> | **CWR + MT$^3$** | 60.58       |
>
> The experimental results show that our two proposed components jointly contribute to improved overall performance. Our CWR component (CWR + MT$^3$) improves over QB (QB +  MT$^3$) by 4.81 points. This further demonstrates that, in few-shot setting, both retrieving more informative demonstrations and adopting the meta test-time training (TTT) paradigm are crucial for enabling the model to perform more effective reasoning. We will incorporate this ablation into the paper.
>
> **W4. [There are no ablation studies examining the chain-of-thought prompt.]**
>
> **A:** Thanks for the insightful review. To analyze the effectiveness of different CoT prompt types, we conduct an ablation study on the MathVista (GPS) benchmark under the same MT$^3$ training framework. Specifially, we compare: (1) the original zero-shot prompt (as shown in Figure 9 of the paper), (2) the general Retrieval-Specific CoT prompt (as shown in Q2 and Figure 12 of the paper), and (3) the Retrieval-Specific CoT prompt tailored for mathematics (as shown in Figure 3 of the paper). All other components are kept fixed, where we train with MT$^3$. The results are as follows:
>
> **Table: Ablation of different CoT prompt of CoT-MT$^3$ on MathVista (GPS)**
>
> | CoT Prompt Type                  | 2-shot    | 4-shot    | 6-shot    |
> | -------------------------------- | --------- | --------- | --------- |
> | W/o CoT (Query-Based Retrieval)  | 50.96     | 55.77     | 56.73     |
> | Zero-shot CoT                    | 55.77     | 56.73     | 56.25     |
> | Retrieval-Specific CoT (General) | 56.25     | 59.62     | 57.21     |
> | Retrieval-Specific CoT (Math)    | **57.21** | **60.58** | **58.65** |
>
> We observe three consistent trends:
>
> - **CoT vs. no CoT.** Introducing any CoT already yields a large improvement over “W/o CoT (Query-Based Retrieval)”, showing that incorporating explicit reasoning is important for effective retrieval.
> - **Retrieval-specific vs. Zero-shot CoT.** The general retrieval-specific CoT substantially outperforms the standard zero-shot CoT (e.g., 4-shot: 56.73 → 59.62), suggesting that a CoT explictly designed for retrieval brings extra benefits over a purely solution-oriented CoT.
> - **Task-tailored retrieval-specific CoT design.** The math retrieval-specific CoT further improves over the general one, suggesting that retrieval-specific CoT design can be task-tailored which leads to consistently better retrieval and reasoning performance.
>
> These results support that our retrieval-specific CoT design is both effective and well justified. We will incorporate this ablation into the paper.

---

> ### Author Response · Authors · 2025-11-24
> **Authors' Response (3/3)**
>
> **Q1. [Do you think that the approach could benefit from a per-sample adaptive `w` argument, where `w` takes a different value depending on the question?]**
>
> **A:** Thanks. Our method could benefit from an adaptive weight $w$. While $w$ is originally introduced to balance the contribution of the test query and the generated CoT cues in retrieval. This balance could be further refined across different samples.
>
> For example, in some extreme cases, the original question may induce noisy retrieval, whereas the model-generated CoT cues offer a cleaner reformulation; in such cases, $w$ should be reduced. In other cases, the question itself is very informative and a larger $w$ would be more appropriate. However, accurately assessing the information contained in each query and the corresponding CoT would require a additional reward model and additional computation, which we leave for future work.
>
> In the revision, we also add visualizations of retrieval demonstrations under different values of $w$ for our CWR strategy in Appendix B.8, to provide a clearer picture of how this weighting affects the actual retrieval.
>
> **Q2. [Could you also please share the exact prompt used for GQA evalautions?]**
>
> **A:** Thank you for your point. We provide the exact prompt used for GQA and M$^3$CoT in Appendix B.4. The prompt used for GQA is as follows:
>
> ```text
> You are an expert in visual question answering. I will now provide you with a multimodal problem.
> Your task is to:
> **1. Understand the problem and list the information:**
> -List all the given information and elements from the text and the image in the problem.
> **2. Identify the key information and unknowns:**
> -Identify critical information for solving the problem and highlight any unknowns that need to be determined.
> **3. Reason step-by-step based on your understanding:**
> -Based on your understanding of the problem, attempt to break it down into logical steps and provide a step-by-step reasoning approach to solving the problem.
> The problem you need to solve is:
> <image>
> <question>
> ```

---

> ### Author Response · Authors · 2025-11-27
> **Authors' Response to Reviewer wwJS**
>
> Thank you once again for taking the time to review our paper and for providing such insightful and constructive feedback. We are currently running the additional experiments as you suggested and will soon update our response with the new results.

---

### Author Response · Authors · 2025-12-03
**Summary of Rebuttal**

Dear AC, SAC, and PC,

We sincerely thank the AC, SAC, PC, and reviewers for your time and effort in handling our submission and providing constructive feedback to improve our work.

After the rebuttal, **our detailed responses have led reviewers m2RX and 88T1 to raise the score to 6**, explicitly confirming that all their concerns have been addressed. Moreover, Reviewers CNNH and wwJS acknowledged the significance of our work, noting that it focuses on "**a currently active and impactful direction**" and explores "**an open and under-researched area**". Their **latest comments emphasized the interest in our work**, focused on clarifications rather than new fundamental concerns, **indicating a positive view**.

To facilitate your final assessment, we summarize the main contributions of our work and the key outcomes of rebuttal as follows:

---

### **Part I: Main Contributions of Our Work**

Our work focuses on few-shot multimodal reasoning, especially complex reasoning, which remains one of the most challenging settings. We have successfully addressed several challenges that were either overlooked or difficult to resolve in previous works:

**i) Addressing Unreliable Retrieval**: We propose CoT-guided Weighted Retrieval (**CWR**), highlighting key information of test query for accurate retrieval by leveraging a retrieval-specific CoT.

**ii) Maximizing Demonstration Utility**: We introduce a Meta Test-time Training (**MT**$^3$) paradigm that constructs dynamic training samples, enabling the model to fully exploit the information within demonstrations.

By combining CWR and MT$^3$, our proposed **CoT-MT**$^3$ retrieves more reasoning-relevant demonstrations and leverages them effectively, yielding robust reasoning performance across different few-shot settings.

---

### **Part II: Key Outcomes of Rebuttal**

1. **Reviewer wwJS (Rating: 4)**

   - **[Review]** Raised questions regarding generalization across different models/domains and CoT prompt ablations.
   - **[Response]**
     - **Generalization**: We have validated the effectiveness consistently across different models/domains, demonstrating **effective transfer to broader multi-modal settings**.
     - **CoT Prompt Ablation**: We have added an ablation over **different CoT prompt types**. Following the reviewer's follow-up clarification, we have also added a **per-subtask CoT prompt ablation** and clarified the model versions.

   - **[Outcome]** We believe that these additional results **directly address the reviewer’s follow-up questions**. The reviewer's latest comments focused on these clarifications instead of new fundamental concerns, **indicating a positive view**.

2. **Reviewer m2RX (Rating: 4 → 6)**

   - **[Review]** Raised concerns regarding direct evaluation of retrieval quality, generalization, and novelty compared to IDS.
   - **[Response]**
     - **Retrieval Quality**: We compare retrieval strategies in a pure ICL setting, demonstrating that **CWR consistently retrieves demonstrations that are more useful for reasoning**.
     - **Novelty**: We have clarified distinction from IDS and provided CoT prompt ablation justifying our design.
   - **[Outcome]** The reviewer confirmed that **all concerns have been addressed** and raised the score to 6.

3. **Reviewer 88T1 (Rating: 4 → 6)**

   - **[Review]** Raised questions about reporting statistical significance and latency.
   - **[Response]** We have verified improvements via statistical significance and provided a detailed latency analysis to show a **balanced accuracy-latency trade-off**, which is valuable for **accuracy-critical applications**.
   - **[Outcome]** The reviewer confirmed **all concerns addressed** and stated to "increase my score to 6".

4. **Reviewer CNNH (Rating: 4)**

   - **[Review]**
     - **Phase 1**: Raised broad questions covering retrieval process, latency, and other implementation details .
     - **Phase 2**: In the follow-up discussion phase, the focus shifted to prove **whether Retrieval-Specific CoT improves retrieval quality, and consequently downstream reasoning**.

   - **[Response]**
     - **Systematic Clarifications**: In phase 1, we have provided the **requested comprehensive experiments and analysis**, effectively resolving the concerns.
     - **Retrieval Quality**: For Phase 2, we have compared different retrieval strategies in a pure ICL setting. Although the discussion is cut short, we believe that this evidence **fully addresses the follow-up question** (similar to reviewer m2RX’s) and confirms that **retrieval-specific CoT indeed improves retrieval behavior and consequently reasoning performance**.

   - **[Outcome]** In the latest comments, the reviewer explicitly stated that the work is **"indeed very interesting."** We believe our comprehensive responses **directly resolve the reviewer's concerns**, and reviewer's constructive engagement **indicates a positive view** of our work.

Best,\
Authors

---

### Meta-Review · Area_Chair_6hFe · 2026-01-04

**Summary:**

This article constructs CoT through multimodal retrieval to achieve multimodal reasoning.
The reviewers expressed affirmation in the following aspects:
1. The writing is clear and well-structured.
2. The experimental analysis is relatively comprehensive.
3. The method is simple yet effective.

The main concerns raised by the reviewers are as follows:
1. The experimental contributions may be insufficient. Missing experiments include model families/model parameter sizes and ablation studies.
2. The reviewers expressed concerns regarding the originality of the proposed method and the effectiveness of its components.
3. Multiple reviewers noted that the paper has only been validated on mathematical datasets, while it claims to be a unified multimodal approach, which poses a risk of overstatement.

**Reviewer Concerns:**

The author added extensive experiments, including more models and parameter sizes, addressing concerns about the method's generalizability.
However, reviewers still have reservations regarding the effectiveness of the components and the significance of the experimental results. As a result, while some reviewers responded, they did not raise their scores.

**Reviewer Scores:**

Reviewers m2RX and 88T1, after thorough communication, have expressed their willingness to raise the score to 6. However, wwJS and CNNH, after discussion, have maintained their negative evaluations.

---

### Decision · Program_Chairs · 2026-01-26

Reject